# Mass of different snow crystal shapes derived from fall speed measurements

Sandra Vázquez-Martín[1], Thomas Kuhn[1], and Salomon Eliasson[2]

[1]Luleå University of Technology (LTU). Department of Computer Science, Electrical and Space Engineering. Division of Space Technology, 98 128, Kiruna, Sweden.
[2]Swedish Meteorological and Hydrological Institute (SMHI), 601 76, Norrköping, Sweden.

**Correspondence:** Thomas Kuhn (thomas.kuhn@ltu.se)

**Abstract.**

Meteorological forecast and climate models require good knowledge of the microphysical properties of hydrometeors and the atmospheric snow and ice crystals in clouds. For instance, their size, cross-sectional area, shape, mass, and fall speed. Especially shape is an important parameter in that it strongly affects the scattering properties of ice particles, and consequently their response to remote sensing techniques. The fall speed and mass of ice particles are other important parameters both for numerical forecast models and for the representation of snow and ice clouds in climate models. In the case of fall speed, it is responsible for the rate of removal of ice from these models. The particle mass is a key quantity that connects the cloud microphysical properties to radiative properties. Using an empirical relationship between the dimensionless Reynolds and Best numbers, fall speed and mass can be derived from each other if particle size and cross-sectional area are also known.

In this study, ground-based in-situ measurements of snow particle microphysical properties are used to analyse mass as a function of shape and the other properties particle size, cross-sectional area, and fall speed. The measurements for this study were done in Kiruna, Sweden during snowfall seasons of 2014 to 2019 and using the ground-based in-situ instrument Dual Ice Crystal Imager (D-ICI), which takes high-resolution side- and top-view images of natural hydrometeors. From these images, particle size (maximum dimension), cross-sectional area, and fall speed of individual particles are determined. The particles are shape classified according to the scheme presented in our previous study, in which particles sort into 15 different shape groups depending on their shape and morphology. Particle masses of individual ice particles are estimated from measured particle size, cross-sectional area, and fall speed. The selected dataset covers sizes from about 0.1 mm to 3.2 mm, fall speeds from 0.1 m s$^{-1}$ to 1.6 m s$^{-1}$, and masses from 0.2 μg to 450 μg. In our previous study, the fall speed relationships between particle size and cross-sectional area were studied. In this study, the same dataset is used to determine the particle mass, and consequently, the mass relationships between particle size, cross-sectional area, and fall speed are studied for these 15 shape groups. Furthermore, the mass relationships presented in this study are compared with the previous studies.

The resulting mass–size relationships indicate that for certain shapes, in particular columns and related shapes, maximum dimension is not suitable to describe the size of snow particles when determining the Reynolds number. Consequently, mass derived from fall speed for these shapes is not reliable. A closure study done on a selection of simple columns, for which mass is determined geometrically, shows that for this shape a characteristic length, similar to the diameter of the basal facet, is

superior to the maximum dimension, which is similar to the column length, as size parameter. Using a modified Best number, the Best number reduced as a function of area ratio, resulted in even better agreement in the closure study, confirming that the modified Best number approach adopted in this study represents an improvement for columns.

**Keywords:** Natural snow crystals; hydrometeors; microphysical properties; fall speed; mass; ground-based in-situ measurements.

## 1 Introduction

Atmospheric models need accurate knowledge of atmospheric ice crystals and snow particles' microphysical properties to ensure realistic parameterizations (e.g., Stoelinga et al., 2003; Tao et al., 2003). These properties, including size, cross-sectional area, shape, fall speed, and mass of ice particles, cannot be measured directly with remote sensing methods. Therefore, retrieval methods of cloud and snow properties also rely on good assumptions of the microphysical properties.

Particle shape is an essential parameter for retrievals of cloud properties from optical remote sensing (see, e.g., Yang et al., 2008; Baum et al., 2011; Xie et al., 2011; Loeb et al., 2018). Furthermore, it can affect retrievals from active and passive microwave measurements of clouds and snowfall (e.g., Sun et al., 2011; Matrosov et al., 2012; Marchand et al., 2013; Kneifel et al., 2010; Cooper et al., 2017). Therefore, the shape dependence of the other microphysical properties is crucial to ensure accurate parameterizations. The fall speed of ice and snow crystals is a critical parameter for the modelling of the microphysical precipitation processes (Schefold et al., 2002) and the climate as it influences the lifetime of cirrus clouds, the vertical transport of water, and the snowfall rate (e.g., Mitchell et al., 2008). Ice particle mass parameterizations are required to derive ice water content (IWC). IWC is a crucial parameter to describe cloud contribution to the atmospheric models' radiation budget (Waliser et al., 2009; Thornberry et al., 2017).

Therefore, it is desirable to have datasets of falling snow particles based on simultaneous measurements of the microphysical properties maximum dimension (particle size), cross-sectional area, shape, fall speed, and particle mass. If not available as measurement, particle mass or fall speed is retrievable based on all other properties. The fall speed cannot be computed directly from maximum dimension, cross-sectional area, and mass, because the drag force on the particle depends on the drag coefficient $C_D$ that also depends on the fall speed. The dimensionless Best number $X$ that only depends on maximum dimension, cross-sectional area, and mass can eliminate this interdependency. The Best number can then help determine the Reynolds number, Re, through empirical relationships between Re and $X$. Finally, Re is used to calculate the fall speed, $v$.

For spherical particles, this Re–$X$ relationship is well known (Abraham, 1970). Böhm (1989) suggested a modified Re–$X$ relationship to determine $v$ for all snow particles. Mitchell (1996) used that relationship to derive $v$ vs maximum dimension power laws from dimensional power laws of cross-sectional area and mass. Heymsfield and Westbrook (2010) suggested a shape-dependent modification of the Best number based on the area ratio. With this modified Best number, they showed that

the error in fall speed determined from the $\mathrm{Re}$–$X$ relationship could be reduced for particles with open geometries, i.e. particles with low area ratio.

Instead of deriving fall speed from mass, the $\mathrm{Re}$–$X$ relationship may also be used to determine mass from measured fall speed. The Reynolds number can be derived from fall speed, and then mass from $X$ together with maximum dimension and cross-sectional area. Szyrmer and Zawadzki (2010) have done this to determine average $v$ vs mass relationships from measurements of snow aggregates' fall speeds.

In this study, the $\mathrm{Re}$–$X$ relationship together with the modified Best number (Heymsfield and Westbrook, 2010) is used to determine masses of individual particles from measured maximum dimensions, cross-sectional areas, and fall speeds given by the dataset of our previous study, Vázquez-Martín et al. (2021), that also includes particle shape. We analyse mass relationships as functions of maximum dimension, cross-sectional area, and fall speed for different snow particle shapes. Section 2 describes the dataset used in this study. Section 3 shows the derivation of particle mass and mass relationships. Section 4 shows and discusses the resulting relationship between mass, size, cross-sectional area, and fall speed. All relationships are studied separately for various particle shapes. In the same section, we also present comparisons between our mass relationships and those from previous studies. In Sect. 5, this study is summarized and concluded.

## 2 Dataset

The dataset consists of 2461 high-resolution dual images of falling natural snow crystals and other hydrometeors. The same dataset has been used in Vázquez-Martín et al. (2021). The data have been collected using D-ICI, the ground-based in-situ instrument described in Kuhn and Vázquez-Martín (2020), at a site in Kiruna, Sweden (67.83° N, 20.41° E), described in Vázquez-Martín et al. (2020) during multiple snowfall seasons, the winters of 2014/2015 to 2018/2019. The images are taken when the snow particles fall into the inlet and consequently fall down the sampling tube and traverse the optical cell. In the centre of the optical cell is the sensing volume. If particles are falling through the sensing volume they are detected by the detecting optics (for a detailed description see Kuhn and Vázquez-Martín, 2020). Upon detection, the particles are optically imaged simultaneously from two different viewing directions. One is horizontal, recording a side view, and one is close to vertical, recording a top view. From the top-view images, we can determine for each particle its maximum dimension $D_{\mathrm{max}}$, which we use to describe particle size, cross-sectional area $A$, and area ratio. From the side-view images, since they are exposed twice, we can determine fall speed. These images are high-resolution (optical resolution of about 10 µm) where one pixel corresponds to 3.7 µm. The additional information dual images provide, improves the shape classification carried out by looking at both top- and side-view images. The particles are classified according to their shape and sorted into 15 different shape groups as described in Vázquez-Martín et al. (2020). A complete description of the dataset and data processing methods is given by Vázquez-Martín et al. (2021).

## 3 Methods

### 3.1 Mass derivation

The motion of hydrometeors when free-falling through the atmosphere establishes an equilibrium between two forces; the gravity and the aerodynamic drag. The resulting particle settling speed is called fall speed $v$. Thus, the fall speed is governed by the physical properties of the hydrometeors, including their mass and projected area, and it involves aerodynamic principles and environmental conditions. The gravitational force is proportional to the particle mass $m$, while the frictional or drag force is proportional to both the particle projected area, i.e. the cross-sectional area $A$, and the square of its fall speed $v$. The force balance yields

$$m \cdot g = \frac{1}{2} \cdot \rho_a \cdot v^2 \cdot A \cdot C_D, \tag{1}$$

where $g$ is the gravitational acceleration, $\rho_a$ the air density, and $C_D$ the drag coefficient. To determine $v$ from the particle properties $m$ and $A$ using this equation, the drag coefficient $C_D$ has to be known as well. However, $C_D$ depends on maximum dimension, shape, and on $v$ itself. To circumvent these interdependencies, one can first determine the Best number $X = C_D \cdot \text{Re}^2$ by rearranging Eq. 1 together with the Reynolds number

$$\text{Re} = \frac{\rho_a \cdot v \cdot D_{\max}}{\eta}, \tag{2}$$

where $\eta$ is the dynamic viscosity of air, to get an expression that does not depend on fall speed $v$:

$$X = \frac{2 \cdot m \cdot g \cdot \rho_a \cdot D_{\max}^2}{A \cdot \eta^2}. \tag{3}$$

Thus, $X$ can be calculated from the particle properties $D_{\max}$, $A$, and $m$. If the relationship between $\text{Re}$ and $X$ is known, one can determine $\text{Re}$ from $X$. In these circumstances, Eq. 2 provides the fallspeed, $v$. Böhm (1989) provides a relationship between $\text{Re}$ and $X$ for snow particles, which is shown here in the form given by Mitchell (1996)

$$\text{Re} = \frac{\delta_0^2}{4} \cdot \left[ \left( 1 + \frac{4 \cdot X^{1/2}}{\delta_0^2 \cdot C_0^{1/2}} \right)^{1/2} - 1 \right]^2, \tag{4}$$

where $\delta_0$ and $C_0$ are unit-less constants, and uses it together with the approach described above to determine $v$ from the particle properties $D_{\max}$, $A$, and $m$.

In a similar approach, one can determine particle mass if $D_{\max}$, $A$, and $v$ are known. For this, $\text{Re}$ is determined from $v$ and $D_{\max}$ using Eq. 2. Then, $X$ is determined from Eq. 4 solved for $X$

$$X = \frac{\delta_0^4 \cdot C_0}{16} \cdot \left\{ \left[ \left( \frac{4 \cdot \text{Re}}{\delta_0^2} \right)^{1/2} + 1 \right]^2 - 1 \right\}^2. \tag{5}$$

Finally, $m$ is added to the dataset using Eq. 3

$$m = \frac{X \cdot A \cdot \eta^2}{2 \cdot g \cdot \rho_a \cdot D_{\max}^2}, \tag{6}$$

where the atmospheric conditions can be accounted for each particle by adapting $\eta$ and $\rho_{\mathrm{a}}$ to the measured temperature and pressure.

Instead of using Eq. 4 or Eq. 5 with one set of $\delta_0$ and $C_0$ for all particles regardless of their shape, as proposed by Böhm (1989), Heymsfield and Westbrook (2010) suggested using a modified Best number $X^*$, replacing $X$ in Eq. 4 or Eq. 5, to correct for effects due to open-geometry shapes. They proposed $X^* = X \cdot A_{\mathrm{r}}^{1/2}$, where $A_{\mathrm{r}} = \frac{A}{\frac{\pi}{4} \cdot D_{\mathrm{max}}^2}$ is the area ratio, which is close to 1 for compact shapes and smaller the more open the geometry is. Heymsfield and Westbrook (2010) showed that by using this approach they could reduce errors of determined fall speeds associated to open-geometry particles with low area ratios. Using our data for simple thick columns in shape group *(3)*, we could confirm that their approach is better than the approach by Böhm (1989) without modifying $X$ (see Appendix C). Therefore, here, we use the modified Best number $X^*$. Consequently, Eq. 6 is modified to

$$m = \frac{X^*}{A_{\mathrm{r}}^{1/2}} \cdot \frac{A \cdot \eta^2}{2 \cdot g \cdot \rho_{\mathrm{a}} \cdot D_{\mathrm{max}}^2} = \frac{\pi \cdot \eta^2 \cdot X^* \cdot A_{\mathrm{r}}^{1/2}}{8 \cdot g \cdot \rho_{\mathrm{a}}}. \tag{7}$$

Note, that then the Best number determined from Eq. 5 is the modified Best number $X^*$. In Eq. 5, we use $\delta_0 = 8.0$ and $C_0 = 0.35$ from Heymsfield and Westbrook (2010).

## 3.2 Fitting relationships to data

Once mass is calculated, we can parameterize the relationships mass vs maximum dimension, $m(D_{\mathrm{max}})$, mass vs cross-sectional area, $m(A)$, and fall speed vs mass, $v(m)$, by fitting the following power laws to our data:

$$m(D_{\mathrm{max}}) = \tilde{a}_D \cdot \left( \frac{D_{\mathrm{max}}}{1 \text{ mm}} \right)^{\tilde{b}_D}, \tag{8}$$

$$m(A) = \tilde{a}_A \cdot \left( \frac{A}{1 \text{ mm}^2} \right)^{\tilde{b}_A}, \tag{9}$$

$$v(m) = a_m \cdot \left( \frac{m}{1 \text{ μg}} \right)^{b_m}, \tag{10}$$

which represent straight lines on logarithmic plots. Hence, linear least-squares fits to the logarithm of the data yield the parameters $\tilde{a}_D$, $\tilde{b}_D$, $\tilde{a}_A$, $\tilde{b}_A$, $a_m$, and $b_m$. The parameter $\tilde{a}_D$ corresponds to the mass at $D_{\mathrm{max}} = 1$ mm, $\tilde{a}_A$ to the mass at $A = 1$ mm$^2$, and $a_m$ to the fall speed at $m = 1$ μg. The parameters $\tilde{b}_D$, $\tilde{b}_A$, and $b_m$ are the exponents in the power laws and the slopes in the linear fits.

As seen in Vázquez-Martín et al. (2021), using binned data instead of individual data reduces the data spread so that fit-functions based on binned data are more robust than fit-functions based on individual data. Therefore, also here the data are first binned into a suitable number of bins before fitting Eq. 8–Eq. 10 to the data. Ten mass bins (for $m$ vs $D_{\mathrm{max}}$ and $m$ vs $A$ relationships) and ten fall speed bins (for $v$ vs $m$) are used, respectively. The bins are spaced such that each bin contains as close to the same number of particles as possible. As a consequence, the bin widths are variable and specific to each shape group, and thereby avoid the problem of individual bins having a disproportional effect on the fit. The binned data consist of the

145 median values for each bin. Then, the $m$ vs $D_{\mathrm{max}}$, $m$ vs $A$, and $v$ vs $m$ relationships are fitted to the median masses vs median maximum dimensions, median masses vs median cross-sectional areas, and median fall speeds vs median masses, respectively. Vázquez-Martín et al. (2021) found that about 40 particles in a shape group (currently the lowest number in our dataset is 37) is the limit where binning can still be used. The advantages of binning become prominent only at larger numbers of particles.

### 3.3  Analytical derivation of relationships

These relationships may be useful for parameterizations in models and retrievals and are readily comparable to other studies. In case a suitable dataset is not available, an alternative to fitting these relationships to measured data, is to derive particle mass analytically from previously determined parameterizations of cross-sectional area vs maximum dimension ($A$ vs $D_{\mathrm{max}}$), fall speed vs maximum dimension ($v$ vs $D_{\mathrm{max}}$) and fall speed vs cross-sectional area ($v$ vs $A$) given by power laws

$$A(D_{\mathrm{max}}) = a \cdot \left( \frac{D_{\mathrm{max}}}{1\ \mathrm{mm}} \right)^{b}, \tag{11a}$$

$$D_{\mathrm{max}}(A) = a' \cdot \left( \frac{A}{1\ \mathrm{mm}^2} \right)^{b'}, \tag{11b}$$

$$v(D_{\mathrm{max}}) = a_D \cdot \left( \frac{D_{\mathrm{max}}}{1\ \mathrm{mm}} \right)^{b_D}, \tag{12a}$$

$$D_{\mathrm{max}}(v) = a'_D \cdot \left( \frac{v}{1\ \mathrm{m\ s^{-1}}} \right)^{b'_D}, \tag{12b}$$

$$v(A) = a_A \cdot \left( \frac{A}{1\ \mathrm{mm}^2} \right)^{b_A}, \tag{13a}$$

$$A(v) = a'_A \cdot \left( \frac{v}{1\ \mathrm{m\ s^{-1}}} \right)^{b'_A}. \tag{13b}$$

For each relationship, the inverse is also shown as the corresponding parameters are convenient for some of the derivations. The parameter $a$ corresponds to the cross-sectional area at $D_{\mathrm{max}} = 1$ mm, $a'$ corresponds to the maximum dimension at $A = $

$1\ \mathrm{mm}^2$, $a_D$ to the fall speed at $D_{\mathrm{max}} = 1$ mm, $a'_D$ to the maximum dimension at $v = 1$ m s$^{-1}$, $a_A$ to the fall speed at $A = 1\ \mathrm{mm}^2$, and $a'_A$ to the cross-sectional area at $v = 1$ m s$^{-1}$. The parameters $b$, $b'$, $b_D$, $b'_D$, $b_A$, and $b'_A$ are the exponents in the power laws.

The resulting power laws are

$$m(D_{\mathrm{max}}) = \frac{\pi^{1/2} \cdot \eta^2 \cdot \gamma}{4 \cdot g \cdot \rho_{\mathrm{a}}} \cdot \left( \frac{a}{1\ \mathrm{mm}^2} \right)^{1/2} \cdot \left( \frac{a_D \cdot \rho_{\mathrm{a}} \cdot 1\ \mathrm{mm}}{\eta} \right)^{\delta} \cdot \left( \frac{D_{\mathrm{max}}}{1\ \mathrm{mm}} \right)^{b_D \cdot \delta + \delta + \frac{1}{2} \cdot b - 1} \tag{14}$$

$$m(A) = \frac{\pi^{1/2} \cdot \eta^2 \cdot \gamma}{4 \cdot g \cdot \rho_\mathrm{a}} \cdot \frac{1 \text{ mm}}{a'} \cdot \left( \frac{a_A \cdot a' \cdot \rho_\mathrm{a}}{\eta} \right)^\delta \cdot \left( \frac{A}{1 \text{ mm}^2} \right)^{b_A \cdot \delta + b' \cdot \delta + 1/2 - b'}, \tag{15}$$

$$v(m) = 1 \text{ m s}^{-1} \cdot \left[ \frac{4 \cdot g \cdot \rho_\mathrm{a} \cdot a_D' \cdot 1 \text{ µg}}{\pi^{1/2} \cdot \eta^2 \cdot \gamma \cdot a_A'^{1/2}} \cdot \left( a_D' \cdot 1 \text{ m s}^{-1} \cdot \frac{\rho_\mathrm{a}}{\eta} \right)^{-\delta} \right]^{\frac{1}{b_D' \cdot (\delta - \frac{1}{2}) + \frac{1}{2} \cdot b_A'}} \cdot \left( \frac{m}{1 \text{ µg}} \right)^{\frac{1}{b_D' \cdot (\delta - \frac{1}{2}) + \frac{1}{2} \cdot b_A'}}. \tag{16}$$

The derivation of these power laws is shown in Appendix B (Eq. B3–Eq. B6). There, also the $X$ vs $\mathrm{Re}$ relationship is expressed as power law instead of using Eq. 5. This can be done by approximating Eq. 5 piece-wise in several regions of $X$ with power laws Eq. B1 (with coefficient $\gamma$ and exponent $\delta$), as done by (Mitchell, 1996). Note, that both methods for deriving the relationships given by Eq. 8–Eq. 10, i.e., either the method described in Sect. 3.1 with fitting detailed in Sect. 3.2 or the alternative derivation from existing relationships described in this section, are equivalent if they are based on the same dataset. The two methods will yield the same relationships if both use the same power law approximations of $X$ vs $\mathrm{Re}$ and the same atmospheric conditions (given as constant $\eta$ and $\rho_\mathrm{a}$ for the whole dataset). Thus, in this study, we have chosen to fit Eq. 8– Eq. 10 directly to our data (Sect. 3.2). This allows using environmental conditions individually for each particle and avoids the need to consider error propagation when deriving new relationships from existing ones.

## 4 Results and discussions

### 4.1 Results from fitting and correlations

The particle masses have been determined from measured $D_{\max}$, $A$, and $v$ with the method described in Sect. 3.1. The $m$ vs $D_{\max}$, $m$ vs $A$, and $v$ vs $m$ relationships given by Eq. 8–Eq. 10 are then fitted to the resulting data, now consisting of $D_{\max}$, $A$, $v$, and $m$, for the 15 shape groups using the fitting method based on binned data described in Sect. 3.2. Figure 1 and Table 1 show the results. For simplicity, we use short names included in Table 1 for the shape groups from here on, and Fig. 1 shows their full names. The large spread in the data represented as individual points is apparent in Figs. A1–A3 in Appendix A.

When fitting $m$ vs $D_{\max}$, $m$ vs $A$, and $v$ vs $m$ relationships to the binned data, we note that, in general, there is a high correlation ($0.9 \lesssim R^2 \lesssim 1$) for most shape groups. In the following, we call the correlation coefficients $R_D^2$, $R_A^2$, and $R_m^2$ to indicate to which of the three relationships they belong to. For the $m$ vs $D_{\max}$ relationship, the only exceptions to high correlations are shape groups *(1) Needles*, *(2) Crossed needles*, and *(3) Thick columns*, as well as *(6) Stellar* and *(10) Spatial plates*, which both have low number of particles, having all $R_D^2 \simeq 0.7$. For the $m$ vs $A$ relationship, only shape groups *(2)* and *(6)*, with $R_A^2 \simeq 0.8$, and *(10)* with $R_A^2 \simeq 0.5$, have a lower correlation. Only in the case of *(10)*, it is uncertain if the fit function is representative of the measured data, as judged by the low $R_A^2$. In most shape groups, the coefficients $R_D^2$ and $R_A^2$ are similar. Only the four groups *(1)*, *(2)*, *(3)*, and *(10)*, mentioned above with lower correlation in one of the relationships, have a distinct difference between $R_D^2$ and $R_A^2$. Of these, the three shape groups *(1)–(3)* have a better correlation for $m$ vs $A$ than for $m$ vs $D_{\max}$, which is consistent with a better $v$ vs $A$ correlation than $v$ vs $D_{\max}$ for the same groups (Vázquez-Martín et al., 2021), given that we have derived $m$ using measured $v$ here.

For the $v$ vs $m$ relationship, all values of $R_m^2$ are 0.85 or higher. These high values indicate that $v$ is better correlated to $m$ than to $D_{\max}$ or $A$ (see the generally lower $R^2$ values reported in Vázquez-Martín et al., 2021). The generally very high correlations are partly also a consequence of $m$ being derived from $v$, rather than being an independent measurement.

## 4.2 Mass versus $D_{\max}$ and $A$

Figure 1a and Fig. 1b show the $m$ vs $D_{\max}$ and $m$ vs $A$ relationships including, for reference, the mass of liquid water spheres symbolizing rain or fog droplets given by the power laws $m = \frac{\pi}{6} \cdot \rho_w \cdot D_{\max}^3$ and $m = \frac{4 \cdot \rho_w}{3 \cdot \sqrt{\pi}} \cdot A^{3/2}$, respectively, where $\rho_w = 1 \, \mathrm{g\,cm^{-3}}$ is the density of liquid water. The mass of spheres is proportional to $D_{\max}^3$ and to $A^{3/2}$. Thus, comparing to Eq. 8 and Eq. 9, one can see that the exponents $\tilde{b}_D = 3$ and $\tilde{b}_A = 1.5$ for spheres. The values of $\tilde{a}_D$ and $\tilde{a}_A$ for spheres are 524 µg, the mass of a droplet with 1 mm diameter, and 752 µg, the mass of a droplet with a cross-sectional area $A = 1 \, \mathrm{mm}^2$, respectively.

### 4.2.1 Slopes $\tilde{b}_D$ and $\tilde{b}_A$

The exponent $\tilde{b}_D$ for shape groups *(12) Graupel* and *(15) Spherical* is close to the value of 3 for spheres, 2.74 and 2.84, respectively. For the same groups, $\tilde{b}_A$ is close to the value of 1.5 for spheres, 1.34 and 1.43 for shape groups *(12) Graupel* and *(15) Spherical*, respectively. For these shape groups, this is expected due to their spherical or roundish morphology. These exponent values, corresponding to the slopes in Fig. 1a) and b) are among the highest values for all shape groups. Shape groups *(6) Stellar* and *(11) Spatial stellar* are the only other shape groups that have similarly steep $m$ vs $D_{\max}$ and $m$ vs $A$ relationships. These two groups do not have a roundish morphology that could explain this. However, a slope similar to spherical particles may indicate that the morphology remains similar in these groups independent of size, i.e., ice particles scale equally in all three dimensions. An example for this would be hexagonal plates or columns that all have the same aspect ratio. For pristine stellar particles one may not expect such a steep slope similar to spherical particles, but rather a decreasing area ratio with increasing size. Shape group *(6)*, however, contains other shapes besides pristine stellar particles, such as rimed stellar and split stellar crystals. A particular mix of shapes may cause an apparently steep slope. Indeed, the area ratio in this shape group is approximately constant (Vázquez-Martín et al., 2021). Our dataset does not contain a sufficient number of stellar particles yet to analyse this further, by for example regrouping particle shapes. Additionally, the low number of particles in this group also results in a relatively high uncertainty ($\tilde{b}_D = 2.61 \pm 0.59$ and $\tilde{b}_A = 1.34 \pm 0.29$).

For most other shape groups, the exponent $\tilde{b}_D$ varies between 1.2 and 2, and all other $\tilde{b}_A$ values range between 0.8 and 1.2. Three shape groups, *(1) Needles*, *(2) Crossed Needles*, and *(3) Thick columns*, stand out with the lowest exponents $\tilde{b}_D$ of approximately 0.8 or lower. These can easily be seen in Fig. 1a) as the lines with the most shallow slopes. For these groups, this is understandable due to their morphology. We have seen in Vázquez-Martín et al. (2021) that an increase in $D_{\max}$ (needle length) is directly proportional to $A$, indicating that the diameter of these needle-shaped particles (needle width) remains similar, when $D_{\max}$ and consequently also $A$ are growing. Thus, $D_{\max}$ is approximately proportional to $A$, and predictably, both $\tilde{b}_D$ and $\tilde{b}_A$ are close to 1 for these three shape groups. Vázquez-Martín et al. (2021), observing the very poor correlation between $D_{\max}$ and measured fall speed for these shape groups, argued that $D_{\max}$ is not suitable to determine the Reynolds number. Therefore, a more suitable characteristic length than $D_{\max}$ should be used to determine Reynolds number

and derive mass from it. Otherwise, the derived mass, and consequently $\tilde{b}_D$, are likely not useful. Jayaweera (1971) suggested a characteristic length for hexagonal crystals, for which the dimensions of the basal facet and the aspect ratio are known. We have

tested this for a small subset of relatively simple columns from shape group *(3)*. Indeed, this charecteristic length, which can easily be determined from the geometric dimensions on the image, is better suited to calculate Re. If it is used instead of $D_{\max}$ in our method to determine $m$ (see Sect. 3.1 and Sect. 3.2), then $\tilde{b}_D$ is now 2.4 (for this subset of columns). Furthermore, the mass can be estimated directly from the geometric dimensions, which allows a closure study to confirm the Re–$X$ relationship, which is central in our method (see Sect. 3.1). The details of this closure study are reported in Appendix C.

Unfortunately, for many particles the dimensions of the basal facet and the aspect ratio are not readily available (or are not defined in case of more complex particles). Consequently, the characteristic length cannot be used and one may want to use $D_{\max}$ instead. However, as the closure study shows (see Appendix C), for shape groups *(1)–(3)* $D_{\max}$ should not be used as a representative size parameter instead of characteristic length. Thus, mass determined using $D_{\max}$ and relationships based on this mass should not be used.

The ratio between the exponents $\tilde{b}_D$ and $\tilde{b}_A$ is equal to the exponent $b$, as can be seen from Eq. 8, Eq. 9, and Eq. 11a. Figure 2 shows the ratios $\frac{\tilde{b}_D}{\tilde{b}_A}$ plotted vs $b$, and most ratios on this plot are close to the line, $\frac{\tilde{b}_D}{\tilde{b}_A} = b$, and range between 1.7 to 2. The exceptions which have ratios much below the line are the two shape groups with the lowest $R_D^2$, groups *(1) Needles* and *(2) Crossed needles*. The ratios for shape groups *(3) Thick columns*, *(9) Side planes*, and *(13) Ice particles* are found slightly below the line. Of these groups, *(3)* and *(13)* are among the groups showing more uncertainty in the determined relationship,

as indicated in Fig. 1a) by the larger confidence regions around the fits. For group *(9) Side planes*, the uncertainty is smaller and can not explain the lower ratio.

Intuitively, $\tilde{b}_D$, the exponent of the $m$–$D_{\max}$ relationship, should be larger than $b$, the exponent of $A$–$D_{\max}$, as confirmed by literature, such as by Mitchell (1996). For some shape groups, however, $b$ is larger than $\tilde{b}_D$. Not surprisingly, groups *(1)–(3)* that were noticed earlier for the lowest $\tilde{b}_D$ values are among these groups, as well as groups *(9)* and *(10)*. The latter two were

noticed in Vázquez-Martín et al. (2021) with very poor correlations between $D_{\max}$ or $A$ and fall speed. This problem likely indicates that for these shape groups $D_{\max}$ is not suitable as size parameter to calculate Re. For simple thick columns, this is demonstrated in Appendix C. While suitable substitutes exist for regular shapes, such as the characteristic length suggested by Jayaweera (1971), for an arbitrary shape our current image analysis methods cannot determine a similar quantity. Thus, the modified $X^*$ approach according to Heymsfield and Westbrook (2010) remains the best alternative for our study, it lessens the

problem considerably for groups *(1)–(3)*.

### 4.2.2   Coefficients $\tilde{a}_D$ and $\tilde{a}_A$

All relationships but those of shape group *(15) Spherical* form a cluster of lines located in a smaller region in both Figs. 1a) and b). The only relationship found outside this cluster is that of shape group *(15)*, which, if extrapolated towards larger sizes or cross-sectional areas, predicts larger masses than any relationship of the other shape groups. The fit coefficients $\tilde{a}_D$ and $\tilde{a}_A$

reflect this since they predict the mass at the unit reference of 1 mm for $\tilde{a}_D$ and 1 mm$^2$ for $\tilde{a}_A$. These values are much larger for spheres, $\tilde{a}_D = 260$ µg and $\tilde{a}_A = 404$ µg, respectively, than for any other shape group. The second-largest values are for

shape group *(12) Graupel*, $\tilde{a}_D = 53.9$ µg and $\tilde{a}_A = 124$ µg, respectively, all other groups have still much smaller values. The smallest values are found for the five groups *(1–3)*, *(6)*, and *(8)*. Of these, *(1–3)* form the lower edge of the cluster of all $m$ vs $D_\mathrm{max}$ relationships except for group *(15)* (Fig. 1a), which furthermore have the lowest and most distinct slopes mentioned earlier. Similarly, these five groups *(1–3)*, *(6)*, and *(8)* form the lower edge of the cluster of all $m$ vs $A$ relationships except for group *(15)* (Fig. 1b).

As can be seen in Figs. 1a) and b), the power laws for *(15)* are close to the reference lines for liquid droplets, however, predicting somewhat lower masses. These differences may be due to several reasons. While shape group *(15) Spherical* may contain liquid droplets, it also contains ice particles that have a lower bulk density $\rho_\mathrm{ice}$ compared to the bulk density of liquid water $\rho_\mathrm{w}$. Also, the small frozen rain droplets that shape group *(15)* contains, are not perfectly spherical, which leads to overestimating mass if assuming a spherical shape. Furthermore, sizing errors cause an apparent error in fall speeds. Overestimating the size leads effectively to too low $v$, which in turn yields too low derived $m$.

### 4.3 Fall speed versus particle mass

The exponent values $b_m$, i.e. the slopes of the $v$ vs $m$ relationships on Fig. 1c), vary less than the slopes of the $m$ vs $D_\mathrm{max}$ and $m$ vs $A$ relationships, they range only from 0.33 to 0.54. The shape groups with the highest slope values include group *(15)* as well as most of the groups that had the lowest slope values in the $m$ vs $D_\mathrm{max}$ and $m$ vs $A$ relationships, $\tilde{b}_D$ and $\tilde{b}_A$, respectively, i.e. groups *(1–3)* and *(6)*. Rather than the slopes, different speeds at any given mass distinguish the different shapes. This can be seen with the values of $a_m$, representing the fall speed predicted by the relationships at the mass given by the reference unit of 1 µg. However, 1 µg is below the masses usually encountered for most shape groups. Therefore, it is more instructive to evaluate predicted fall speeds closer to the median of masses in the dataset. At a mass of for example 3 µg, the fall speeds vary between 0.14 m s$^{-1}$ and 0.53 m s$^{-1}$ as seen in Fig. 1c). The highest four fall speeds at this mass correspond to shape groups *(15)*, *(13)*, *(3)*, and *(12)*, in order of descending speed. These groups contain the most compact shapes. Contrarily, the group with the lowest speed at 3 µg, shape group *(6)*, features the most open structures.

### 4.4 Comparison with previous studies

The mass vs particle size ($m$ vs $D$) and fall speed vs mass ($v$ vs $m$) relationships of the common shapes plates, dendrites, graupel, and spheres, i.e. for our shape groups *(5) Plates*, *(6) Stellar* (called dendrites in other studies), *(12) Graupel*, and *(15) Spherical*, respectively, are compared to previously published relationships based on measurements of mass of individual particles. The parameterizations of $m$ vs $D$ (see Fig. 3a–c) selected for this comparison are taken from Locatelli and Hobbs (1974) L74, Heymsfield and Kajikawa (1987) H87, Kajikawa (1989) K89, Mitchell (1996) M96, and Erfani and Mitchell (2017) E17 and are listed in Table 2. For comparison with $v$ vs $m$ (see Fig. 3d) of our shape groups *(12) Graupel* and *(15) Spherical*, parameterizations of measurements by L74 (see also Table 2) and measurements of Gunn and Kinzer (1949) G49 have been selected. Relationships from this study are further referred to as VM21 and are taken from Table 1. They have been determined as described in Sect. 3.2. Fig. 3 shows all these relationships. For comparison, a line for speeds determined from

**Table 1.** Mass $vs$ maximum dimension ($m$ $vs$ $D_{max}$), mass $vs$ cross-sectional area ($m$ $vs$ $A$), and fall speed $vs$ mass ($v$ $vs$ $m$) relationships given by Eq. 8–Eq. 10 and fitted to our data for each shape group and all data, i.e. for all the particles regardless of shape. The number of particles $N$, parameters $\tilde{a}_D$, $\tilde{b}_D$, $\tilde{a}_A$, $\tilde{b}_A$, $a_m$, $b_m$ with their respective uncertainties, and the correlation coefficients $R^2$ are shown for each shape group and regardless of shape. The root-mean-square error (RMSE) values of base-10 logarithms of measured $m$ $vs$ predicted $m$ (for $m$ $vs$ $D_{max}$ or $m$ $vs$ $A$), and measured $v$ $vs$ predicted $v$ (for $v$ $vs$ $m$) are also shown to indicate the uncertainty of these power laws.

| Shape groups (1–15) | $N$ | $m$ vs $D_{max}$ | | | | $m$ vs $A$ | | | | $v$ vs $m$ | | | |
|---|---|---|---|---|---|---|---|---|---|---|---|---|---|
| | | $\tilde{a}_D$ [µg] | $\tilde{b}_D$ | $R^2_D$ | RMSE | $\tilde{a}_A$ [µg] | $\tilde{b}_A$ | $R^2_A$ | RMSE | $a_m$ [m s$^{-1}$] | $b_m$ | $R^2_m$ | RMSE |
| (1) Needles | 317 | 6.14 ± 0.09 | 0.73 ± 0.21 | 0.67 | 0.10 | 48.5 ± 0.19 | 1.10 ± 0.09 | 0.96 | 0.05 | 0.16 ± 0.07 | 0.42 ± 0.04 | 0.96 | 0.03 |
| (2) Crossed needles | 66 | 6.77 ± 0.10 | 0.88 ± 0.23 | 0.71 | 0.08 | 60.8 ± 0.38 | 1.34 ± 0.25 | 0.82 | 0.11 | 0.11 ± 0.08 | 0.54 ± 0.03 | 0.98 | 0.02 |
| (3) Thick columns | 103 | 6.64 ± 0.27 | 1.13 ± 0.28 | 0.73 | 0.13 | 53.9 ± 0.44 | 1.11 ± 0.15 | 0.90 | 0.09 | 0.20 ± 0.05 | 0.42 ± 0.04 | 0.94 | 0.03 |
| (4) Capped columns | 189 | 14.7 ± 0.09 | 1.65 ± 0.17 | 0.94 | 0.07 | 44.3 ± 0.11 | 0.93 ± 0.06 | 0.98 | 0.05 | 0.20 ± 0.12 | 0.38 ± 0.05 | 0.89 | 0.05 |
| (5) Plates | 197 | 18.6 ± 0.03 | 1.77 ± 0.05 | 1.00 | 0.02 | 35.9 ± 0.09 | 0.96 ± 0.05 | 0.99 | 0.04 | 0.22 ± 0.08 | 0.33 ± 0.04 | 0.91 | 0.04 |
| (6) Stellar | 43 | 5.63 ± 0.27 | 2.61 ± 0.59 | 0.76 | 0.17 | 23.5 ± 0.17 | 1.34 ± 0.29 | 0.79 | 0.15 | 0.08 ± 0.13 | 0.51 ± 0.05 | 0.95 | 0.04 |
| (7) Bullet rosettes | 41 | 21.5 ± 0.08 | 1.98 ± 0.25 | 0.91 | 0.08 | 63.1 ± 0.18 | 1.06 ± 0.14 | 0.90 | 0.08 | 0.22 ± 0.14 | 0.33 ± 0.05 | 0.89 | 0.03 |
| (8) Branches | 438 | 9.30 ± 0.04 | 1.81 ± 0.10 | 0.98 | 0.04 | 29.4 ± 0.09 | 1.02 ± 0.08 | 0.97 | 0.06 | 0.13 ± 0.07 | 0.43 ± 0.03 | 0.97 | 0.03 |
| (9) Side planes | 350 | 18.2 ± 0.03 | 1.45 ± 0.09 | 0.98 | 0.03 | 41.5 ± 0.05 | 0.88 ± 0.06 | 0.98 | 0.03 | 0.16 ± 0.08 | 0.40 ± 0.03 | 0.98 | 0.02 |
| (10) Spatial plates | 48 | 14.2 ± 0.18 | 1.47 ± 0.37 | 0.73 | 0.11 | 26.2 ± 0.55 | 0.83 ± 0.34 | 0.49 | 0.18 | 0.15 ± 0.17 | 0.48 ± 0.08 | 0.85 | 0.05 |
| (11) Spatial stellar | 185 | 15.4 ± 0.07 | 2.56 ± 0.14 | 0.98 | 0.07 | 56.2 ± 0.06 | 1.43 ± 0.05 | 0.99 | 0.04 | 0.17 ± 0.07 | 0.37 ± 0.02 | 0.97 | 0.03 |
| (12) Graupel | 37 | 53.9 ± 0.11 | 2.74 ± 0.15 | 0.98 | 0.05 | 124 ± 0.20 | 1.34 ± 0.09 | 0.97 | 0.07 | 0.24 ± 0.09 | 0.39 ± 0.04 | 0.94 | 0.04 |
| (13) Ice particles | 60 | 15.3 ± 0.22 | 1.46 ± 0.18 | 0.91 | 0.09 | 71.7 ± 0.39 | 1.02 ± 0.12 | 0.92 | 0.10 | 0.26 ± 0.09 | 0.42 ± 0.07 | 0.87 | 0.06 |
| (14) Irregulars | 346 | 14.1 ± 0.08 | 1.91 ± 0.15 | 0.96 | 0.08 | 43.7 ± 0.18 | 1.09 ± 0.11 | 0.94 | 0.11 | 0.18 ± 0.08 | 0.37 ± 0.03 | 0.96 | 0.04 |
| (15) Spherical | 41 | 260 ± 0.78 | 2.84 ± 0.43 | 0.88 | 0.18 | 404 ± 0.76 | 1.43 ± 0.20 | 0.90 | 0.16 | 0.30 ± 0.03 | 0.51 ± 0.02 | 0.99 | 0.03 |
| **All data** | 2461 | 12.8 ± 0.02 | 1.51 ± 0.04 | 1.00 | 0.02 | 38.3 ± 0.03 | 0.97 ± 0.02 | 1.0 | 0.02 | 0.19 ± 0.03 | 0.34 ± 0.01 | 0.99 | 0.01 |

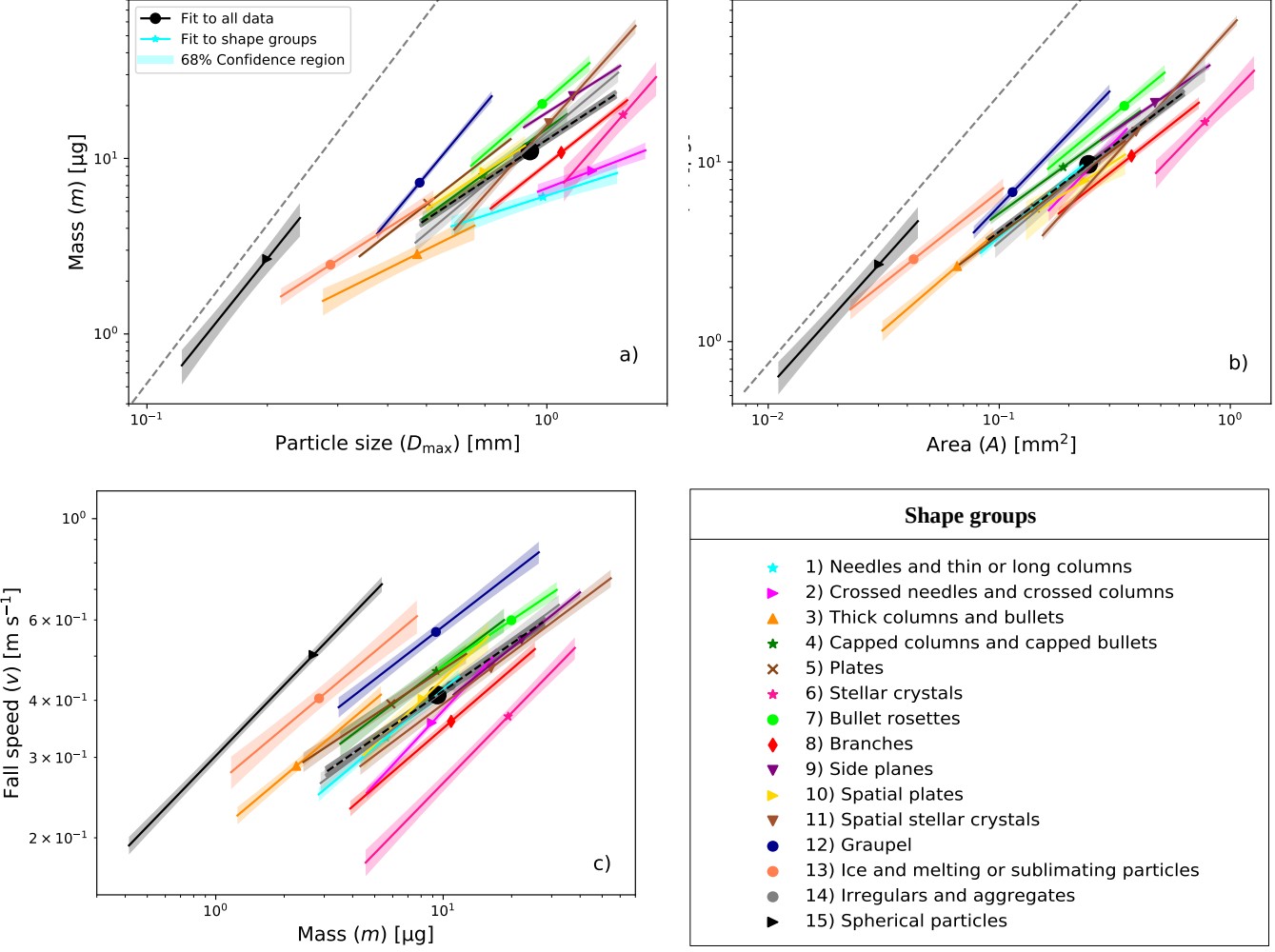

**Figure 1.** Mass vs maximum dimension ($m$ vs $D_{\max}$), mass vs cross-sectional area ($m$ vs $A$), and fall speed vs mass ($v$ vs $m$) relationships are shown in logarithmic scale for all the shape groups (solid lines) and all data (dashed black line). The median $D_{\max}$, $A$, and $m$ of the data is represented as a single point on each line. The length of fit lines is defined by $16^{\text{th}}$ and $84^{\text{th}}$ percentiles of $D_{\max}$, $A$, and $m$. The 68% confidence region for the fits is also shown. **a)** The $m$ vs $D_{\max}$ relationships. For comparison, the mass of spheres, corresponding to rain or fog droplets, given by $m = \frac{\pi}{6} \cdot \rho_{\text{w}} \cdot D_{\max}^3$, where the density $\rho_{\text{w}} = 1\,\text{g cm}^{-3}$, is shown as a grey dashed line. **b)** The $m$ vs $A$ relationships. For comparison, the mass of spheres given by $m = \frac{4 \cdot \rho_{\text{w}}}{3 \cdot \sqrt{\pi}} \cdot A^{3/2}$ is shown as a grey dashed line. **c)** The $v$ vs $m$ relationships.

Eq. 2 using Re calculated from the Re vs $X$ relationship Eq. 4 and $X$ given by Eq. 3 for spherical particles having a density
$\rho_{\text{w}} = 1\,\text{g cm}^{-3}$ is added to the $v$ vs $m$ relationships in Fig. 3. This line will be referred to as [Re–$X$].

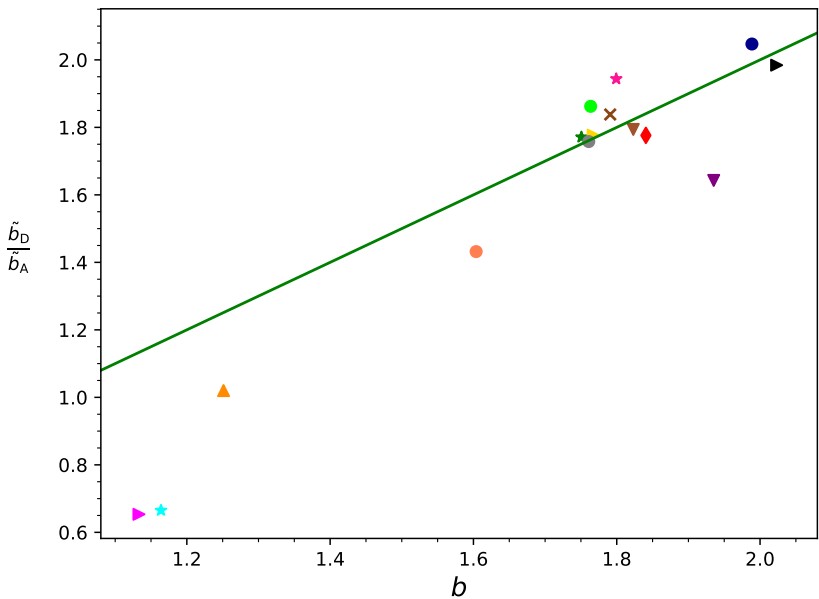

**Figure 2.** Ratio of the exponents $\tilde{b}_{\mathrm{D}}$ and $\tilde{b}_{\mathrm{A}}$ from the $m$ vs $D_{\mathrm{max}}$ and $m$ vs $A$ relationships, respectively, and the exponent $b$ corresponding to $A$ vs $D_{\mathrm{max}}$ relationship are shown for all the shape groups. The green solid line corresponds to the general relationships between the slopes, $\frac{\tilde{b}_{\mathrm{D}}}{\tilde{b}_{\mathrm{A}}} = b$ (derived from Eq. 8, Eq. 9, and Eq. 11a).

Depending on the study, the particle size $D$ was defined somewhat differently. For VM21, as well as for H87, E17, K89, and M96, $D$ corresponds to $D_{\mathrm{max}}$. For L74, $D$ is the diameter of an estimated circle that has the same cross-sectional area as the imaged particle.

### 4.4.1 Plates

We note that for plates (Fig. 3a), the $m$ and $D$ relationship for crystal with sector-like branches (P1b) reported by M96 is most similar to VM21. It is similar also with respect to its slope given by the value of $\tilde{b}_D$, all other relationships are steeper with larger values of $\tilde{b}_D$. Reasonably close to VM21 is also the relationship for hexagonal plate by M96, which, however, is heavier at larger sizes than about 1 mm. For those larger sizes hexagonal plate by M96 is similar to hexagonal plate by H87, the latter having the steepest slope (with $\tilde{b}_D$ even larger than 3). The relationship for P1b by H87 predicts the lightest particles

below about 2 mm. At about 2 mm, it is also similar to VM21 and P1b by M96. Thick plates by H87 are heavier at larger sizes, but similar to VM21 at about 0.2 mm. Our relationship VM21 for shape group *(5)* has a lower slope $\tilde{b}_D$ than any of the other relationships from previous studies. Chen (1992) approximated hexagonal plates with spheroids and found a theoretical lower limit of 2 for $\tilde{b}_D$ of plates, which the value 1.76 of VM21 seems to violate. While the selected previous studies with $\tilde{b}_D$ values larger than 2 looked at particular shapes, VM21's shape group *(5)* represents a mixture of plate-like shapes such

as rimed plates, split plates, and double plates. Two of the shapes are represented with more than 40 particles, namely rimed plates (R1c) and double plates (P1o, see Vázquez-Martín et al., 2020), sufficient to determine their own relationships. As can be seen in Table 2 and Fig. 3a, both have steeper relationships with $\tilde{b}_D$ of approximately 2.1. Double plates are composed of two plates with a small gap in between, so that they almost resemble thicker plates. They are most similar to the thick plates (C1h) by H87 within their size range. Most rimed plates in our dataset are thinner plates with light to moderate riming. They are most similar to hexagonal plates by M96.

### 4.4.2 Dendrites

For stellar particles (Fig. 3b), several $m$ and $D$ relationships are fairly close to VM21, for example the two relationships for P2a from H87, which has a similar slope, and K89, which has a lower slope than VM21. Relationships for another stellar particle type, P1d, are still relatively close to VM21. For example the one by M96 crosses VM21 somewhat above 1 mm and is higher for smaller particles, whereas the one by H87 is about a factor of 2 below VM21. This P1d relationship by H87 may be compared to the rimed stellar (R1d) of the same study H87. These two curves differ by about a factor of two, with the rimed stellar having the larger mass and being very close to VM21. Riming of stellar particles adds mass without increasing their size noticeably (Erfani and Mitchell, 2017) as seen in E17, which explains the difference seen between the two mentioned relationships of H87. A similar difference is seen between the two relationships by E17 from a dataset used to study effects of riming. However, the two relationships by K89, which also feature unrimed and rimed stellar particles, respectively, do not show a significant difference. Particles included in shape group *(6) Stellar* of VM21 include cases of light riming. Distinguishing between unrimed and rimed stellar particles in the data of group *(6)* resulted in two relationships (not shown) that are both, within uncertainties, identical to the one produced from all data in shape group *(6)*.

### 4.4.3 Graupel and spheres

L74 reported three $m$ vs $D$ relationships for lump graupel (R4b) corresponding to three different particle densities with larger masses predicted by the relationships for higher densities. Our relationship for graupel is between L74's low and medium density relationships (Fig. 3c). It is well approximated by the mass of spherical particles with a density of $0.12$ $\mathrm{g\,cm^{-3}}$ (not shown in Fig. 3c), which is at the lower end of the density range reported by L74 for their medium density relationship ($>0.10$ to $0.25$ $\mathrm{g\,cm^{-3}}$). The relationship by H87 for lump graupel (R4b) is similar to L74's medium density. The relationship by E17 agrees also with VM21, but only around 1 mm, as their relationship has a much lower slope ($\tilde{b}_D = 2.16$) than all other relationships for graupel (2.7 to 3.1). The mass of liquid water spheres $m = \frac{\pi}{6} \cdot \rho_{\mathrm{w}} \cdot D^3$ that was shown on Fig. 1a) is added also to Fig. 3c) as reference. Its comparison with VM21's line for shape group *(15) Spherical* is discussed in Sect. 4.2.2.

The $v$ vs $m$ relationships from L74 (Fig. 3d) come, within their ranges, close to our relationship for shape group *(12)*. In general, at a certain particle mass, the size and cross-sectional area, and thus the drag force, decrease with increasing graupel particle density. This can be seen, to some extent, for the three lines by L74. However, their lines have different slopes in a way that makes them intersect with each other. Their slopes are more shallow than the relationship of VM21, consequently they also cross that line. The slope for graupel of VM21 is more similar to that of the relationships related to spherical particles

than the lines for graupel by L74. Consequently it approaches spherical particles, which represent an upper limit in speed, at a lower mass than the lines by L74.

The lines for spherical particles of G49 and [Re–$X$] are very close to each other, thus [Re–$X$] predicts well these measurements. The straight line for the shape group *(15)* of VM21 is at somewhat lower fall speeds below approximately 10 µg. All data but two particles in shape group *(15)* have $m$ below that mass. For those two particles heavier than 10 µg the fit line VM21 over-predicts mass (see Fig. A3 in the Appendix). While VM21 represents the power-law fit to our measurements of droplets and spherical and almost spherical ice particles, the two curved lines of G49 and [Re–$X$] represent only liquid droplets, and,
thus, an upper limit in fall speed.

## 5    Summary and conclusions

This manuscript presents new mass and fall speed parameterizations derived from D-ICI measurements of natural snow, ice crystals and other hydrometeors, covering sizes from 0.1 mm to 3.2 mm. Using the dataset and fall speed vs maximum dimension and vs cross-sectional area relationships from Vázquez-Martín et al. (2021), where fall speeds cover ranges from $0.1\,\mathrm{m\,s^{-1}}$
to $1.6\,\mathrm{m\,s^{-1}}$, in this study, we have added particle masses to our dataset of measured maximum dimension, cross-sectional area, and fall speed of individual particles. The calculated values of individual particle masses range from close to 0.2 µg to 450 µg

Mitchell (1996) presented fall speed relationships derived from power laws of cross-sectional area and mass vs maximum dimension using a relationship between Re and $X$. We calculate particle mass data from our measurements of maximum dimension, cross-sectional area, and fall speed using the same Re–$X$ relationship. With this new extended dataset, mass vs
maximum dimension relationships, mass vs cross-sectional area, and fall speed vs mass, given by Eq. 8–Eq. 10, have been derived and studied for different particle shapes. We present the conclusions that our results led to below.

- As seen in Figs. A1–A3 in Appendix A, and discussed in Section 4.1, the data's large spread is apparent. However, when fitting $m$ vs $D_{\mathrm{max}}$, $m$ vs $A$, and $v$ vs $m$ relationships to binned data, there are high correlation coefficients for most shape groups, with values between 0.9 and 1. The only exceptions are shape groups *(1) Needles*, *(2) Crossed needles*, *(3)*
*Thick columns*, *(6) Stellar*, and *(10) Spatial plates* for the $m$ vs $D_{\mathrm{max}}$ relationship with $R^2_D \simeq 0.7$, as well as for the $m$ vs $A$ relationship shape groups *(2)* and *(6)* with $R^2_A \simeq 0.8$ and *(6)* with $R^2_A \simeq 0.5$. While for all other shape groups $R^2_D$ and $R^2_A$ are similar, for these groups with lower $R^2$, $R^2_D$ is lower than $R^2_A$ for all but shape group *(10)*, for which $R^2_A$ is lower. For $v$ vs $m$, there is a good correlation for all 15 shape groups (see Table 1). The fact that $m$ is derived from $v$ contributes to a stronger correlation between both quantities.

- For the three shape groups related to columnar or elongated shapes, i.e. shape groups *(1)–(3)*, width rather than length or $D_{\mathrm{max}}$ is more closely related to a suitable characteristic length to determine Re (see Sect. 4.2.1 and Appendix C). Consequently, mass and relationships with it are not reliable. For these shape groups, $\tilde{b}_D$ is close to or smaller than 1. Additionally, contrary to expectations $\tilde{b}_D$ is larger than $b$ and the ratio of exponents $\tilde{b}_D$ to $\tilde{b}_A$ is too low for these groups. For most other shape groups it is similar to $b$, as theoretically expected. Shape groups *(9)* and *(10)* (the latter with low

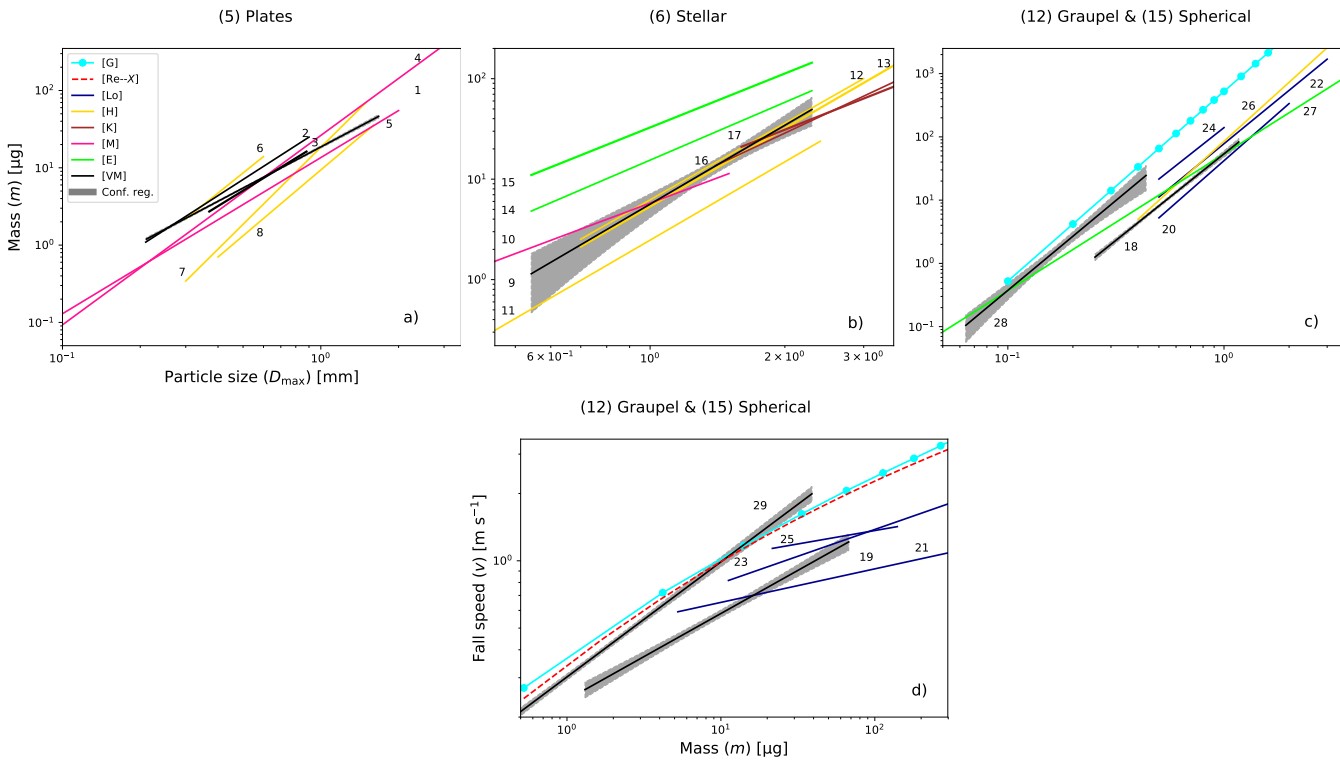

**Figure 3.** A comparison of the mass vs particle size ($m$ vs $D$) and fall speed vs mass ($v$ vs $m$) relationships of this study and previous studies for some shape groups: *(5) Plates*, *(6) Stellar*, *(12) Graupel*, and *(15) Spherical* are shown in logarithmic scale. **a)–c)** The $m$ vs $D$ relationships for *(5) Plates*, *(6) Stellar*, *(12) Graupel* and *(15) Spherical*, respectively. **d)** The $v$ vs $m$ relationships for *(12) Graupel* and *(15) Spherical*. For the comparison, parameterizations from Gunn and Kinzer (1949) *[G]*, Locatelli and Hobbs (1974) *[Lo]*, Heymsfield and Kajikawa (1987) *[H]*, Kajikawa (1989) *[K]*, Mitchell (1996) *[M]*, and this study *[VM]* are shown. In **c)**, the line by *[G]* corresponds to the mass of spheres given by $m = \frac{\pi}{6} \cdot \rho_{\mathrm{w}} \cdot D_{\max}^3$ that was shown also in Fig. 1a). In **d)**, for comparison, a line for speeds determined from Eq. 2 using Re from Eq. 3 and Eq. 4 for spherical particles with density $\rho_{\mathrm{w}} = 1\,\mathrm{g\,cm}^{-3}$ is added as a red dashed line. This line is referred to as [Re–$X$]. These $m$ vs $D$, $v$ vs $m$ relationships are the same shown and enumerated in Table 2. The power laws that correspond to *[VM]* are shown together with their respective 68% confidence regions. The length of all relationships correspond to the ranges of $D_{\max}$ and $m$ in the $x$-axis (see Table 2).

number of particles and low correlations in relationships) show similar limitations when comparing with $b$. Therefore, as long as a more suitable size parameter is not available in our dataset for these shapes, mass derived from Re for these shape groups should only be used with great caution.

- On a selection of 75 simple columns from shape group *(3)*, we have done a closure study (see Appendix C) to confirm the Re–$X$ relationship, which is central in our method (see Sect. 3.1) and used by many other studies. For this, the widths

and lengths of the columns have been determined in addition to $D_{\mathrm{max}}$. From these geometric dimensions, the masses of the columns have been estimated directly. Then, from each column mass, the Best number $X$ has been determined using Eq. 3. Thus, $\mathrm{Re}$ and $X$ have been determined independently and consequently compared to the $X$–$\mathrm{Re}$ relationship given by Eq. 5. This closure showed the superiority of the characteristic length $L^*$ (Jayaweera, 1971) over $D_{\mathrm{max}}$, confirming that $D_{\mathrm{max}}$ is not suitable to calculate $\mathrm{Re}$ and $X$ using Eq. 2 and Eq. 3, respectively, for columns. The closure study also showed that using the modified Best number $X^*$ (Heymsfield and Westbrook, 2010) instead of the Best number $X$ improved the agreement. The best closure for our subset of simple columns was achieved when using both characteristic length $L^*$ and modified Best number $X^*$ together.

- When deriving the $m$ vs $D_{\mathrm{max}}$, $m$ vs $A$, and $v$ vs $m$ relationships analytically from $A$ vs $D_{\mathrm{max}}$, $v$ vs $D_{\mathrm{max}}$, and $v$ vs $A$ given from a suitable dataset (see Section 3.3), the results are equivalent to fitting to the same dataset after adding $m$ for individual particles derived from $v$ (see Sect. 3.1). On the one hand, fitting $m$ vs $D_{\mathrm{max}}$, $m$ vs $A$, and $v$ vs $m$ relationships to data has the advantage that the X–Re relationship from Eq. 5 can be used rather than power-law approximations required for the analytical derivation of the same relationships (see B in Appendix). On the other hand, if a suitable dataset is not available but power-law relationships for $A$ vs $D_{\mathrm{max}}$, $v$ vs $D_{\mathrm{max}}$, and v vs A are, the analytically derived mass relationships Eq. 14–Eq. 16 can be used.

- The parameters $\tilde{b}_D$ and $\tilde{b}_A$, i.e. the slopes of the $m$ vs $D_{\mathrm{max}}$, $m$ vs $A$ power laws, respectively, are highest for the shape groups *(6) Stellar*, *(11) Spatial stellar*, *(12) Graupel*, and *(15) Spherical*. For groups *(12)* and *(15)* they are close to the values expected for spheres, i.e. $\tilde{b}_D = 3$ and $\tilde{b}_A = {}^3/_2$.

- The exponent values $b_m$, i.e. the slopes of $v$ vs $m$, range from 0.33 to 0.55. These $b_m$-slopes do not distinguish the different shapes as seen by the $\tilde{b}_D$-slopes for $m$ vs $D_{\mathrm{max}}$. Instead, different speeds at any given mass are characteristic for the different shapes, with the highest fall speed for *(15) Spherical* and the lowest for *(6) Stellar* that has shapes with open structures.

- We compared our $m$ vs $D_{\mathrm{max}}$ and $v$ vs $m$ relationships with other mass relationships given by previous studies. The shape groups compared in this study are *(5) Plates*, *(6) Stellar*, *(12) Graupel*, and *(15) Spherical*. Our results agree reasonably well with the references used.

- For graupel and spheres, (Section 4.4.3), Locatelli and Hobbs (1974) Lo74 reported $m$ vs $D$ relationships for lump graupel (R4b) with different particle densities (high, medium, and low). Our relationship for graupel is between Lo74's low and medium density relationships, and it is well approximated by the mass of spherical particles with a density of $0.12 \, \mathrm{g \, cm}^{-3}$ (not shown in Fig. 3c).

- Looking at $v$ vs $m$, the two lines for spherical particles of G49 and $[\mathrm{Re}$–$X]$, corresponding to a line for speeds determined from Eq. 2 using $\mathrm{Re}$ from Eq. 3 and Eq. 4 for spherical particles with density $\rho_{\mathrm{w}} = 1 \, \mathrm{g \, cm}^{-3}$, are very close to each other. We report somewhat lower speeds for the shape group *(15) Spherical* VM21. This difference may be due to shape

group *(15)* in VM21 consisting of any spherical or almost spherical particle, including ice, whereas the two lines of G49 and [Re–$X$] are exclusively for liquid droplets.

These resulting parameterizations may improve our understanding of precipitation in cold climates and improve the micro-
physical parameterizations in the climate and forecast models. Through these relationships, we can determine particle masses based on fall speed and particle sizes.

*Data availability.*  The presented data will be available at the Swedish National Data Service (DOI will be available).

*Author contributions.*  Author contributions. TK and SVM performed the conceptualization; TK prepared the resources and the instrumen-
tation; SVM and TK performed the experiments and data collection; SVM and TK prepared the formal analysis; SVM and TK carried out
the data curation; SVM prepared the original draft; SVM, TK and SE contributed to changes and writing during review and revisions; SVM prepared the visualization that includes tables and figures; TK and SE carried out the supervision of the research project.

*Competing interests.*  The authors declare no conflict of interest.

*Acknowledgements.*  We like to thank the Graduate School of Space Technology at the Luleå University of Technology for financial support and the Swedish Institute of Space Physics (IRF) at Kiruna for offering its facilities for our instrument. In addition we thank David L. Mitchell
and an anonymous reviewer for providing valuable feedback and suggestions that helped us to improve our publication.

**Table 2.** The $m$ vs $D$ and $v$ vs $m$ relationships of previous studies given by Locatelli and Hobbs (1974) L74, Heymsfield and Kajikawa (1987) H87, Kajikawa (1989) K89, Mitchell (1996) M96, and Erfani and Mitchell (2017) E17 are shown for some shapes that were selected for the comparison and correspond to *(5) Plates*, *(6) Stellar*, *(12) Graupel*, and *(15) Spherical*. The power laws for M96 have been determined by using equation [15] in Mitchell (1996). The relationships in this study (see Table 1) have been found by fitting Eq. 8–Eq. 10 to our data as described in Sect. 3.2. Those selected for comparison are also shown here as VM21. The snow particles type, the total number of particles $N$, ranges of particle sizes $D$, mass $m$, fall speeds $v$, the $m$ vs $D$, $v$ vs $m$ relationships, the correlation coefficient ($R^2$), and the reference of the studies are displayed. In these references, the particle sizes are defined somewhat differently. In VM21, as well as H87 and M96, $D$ is defined as $D_{\max}$. Magono and Lee (1966)'s symbols are sometimes added for shape clarification. These $m$ vs $D$, $v$ vs $m$ relationships are shown in Fig. 3. The power laws from the literature have been converted in order to have the same units, so that mass $m$ is in µg, particle size $D$ in mm, and fall speed $v$ in m s$^{-1}$.

| Snow particle type | $N$ | Range of $D$ | Range of $m$ | Range of $v$ | Relationships ($m$–$D$, $v$–$m$) | $R^2$ | Ref. |
|---|---|---|---|---|---|---|---|
| **Shape group *(5) Plates*** | 197 | 0.21–1.7 mm | 0.58–57 µg | 0.11–0.9 m s$^{-1}$ | 1. $m\,/(\mathrm{\mu g}) = 18.6 \cdot (D/\mathrm{mm})^{1.77}$ | 1.0 | VM21 |
| Rimed plates (R1c) | 44 | 0.37–0.9 mm | 1.2–17 µg | 0.11–0.6 m s$^{-1}$ | 2. $m\,/(\mathrm{\mu g}) = 21.1 \cdot (D/\mathrm{mm})^{2.06}$ | 0.66 | VM21 |
| Double plates (P1o) | 55 | 0.26–1.5 mm | 1.7–58 µg | 0.21–0.9 m s$^{-1}$ | 3. $m\,/(\mathrm{\mu g}) = 31.3 \cdot (D/\mathrm{mm})^{2.15}$ | 0.88 | VM21 |
| Hexagonal plates | – | 0.10–3.0 mm | – | – | 4. $m\,/(\mathrm{\mu g}) = 26.2 \cdot (D/\mathrm{mm})^{2.45}$ | – | M96 |
| Crystal with sector-like branches (P1b) | – | 0.04–2.0 mm | – | – | 5. $m\,/(\mathrm{\mu g}) = 13.6 \cdot (D/\mathrm{mm})^{2.02}$ | – | M96 |
| Thick plate (C1h) | 19 | 0.30–0.6 mm | 2.6–10 µg | – | 6. $m\,/(\mathrm{\mu g}) = 54.9 \cdot (D/\mathrm{mm})^{2.68}$ | 0.67 | H87 |
| Hexagonal plate (P1a) | 34 | 0.30–1.5 mm | 0.20–70 µg | – | 7. $m\,/(\mathrm{\mu g}) = 18.4 \cdot (D/\mathrm{mm})^{3.31}$ | 0.93 | H87 |
| Crystal with sector-like branches (P1b) | 19 | 0.40–1.6 mm | 0.70–34 µg | – | 8. $m\,/(\mathrm{\mu g}) = 9.38 \cdot (D/\mathrm{mm})^{2.83}$ | 0.97 | H87 |
| **Shape group *(6) Stellar*** | 43 | 0.54–2.3 mm | 1.76–77 µg | 0.13–0.8 m s$^{-1}$ | 9. $m\,/(\mathrm{\mu g}) = 5.63 \cdot (D/\mathrm{mm})^{2.61}$ | 0.76 | VM21 |
| Stellar crystal with broad arms (P1d) | – | 0.09–1.5 mm | – | – | 10. $m\,/(\mathrm{\mu g}) = 5.77 \cdot (D/\mathrm{mm})^{1.67}$ | – | M96 |
| Stellar crystal with broad arms (P1d) | 23 | 0.40–2.4 mm | 0.20–31 µg | – | 11. $m\,/(\mathrm{\mu g}) = 2.47 \cdot (D/\mathrm{mm})^{2.59}$ | 0.95 | H87 |
| Stellar with end plates (P2a) | 11 | 0.70–3.0 mm | 4.9–92 µg | – | 12. $m\,/(\mathrm{\mu g}) = 6.23 \cdot (D/\mathrm{mm})^{2.53}$ | 0.88 | H87 |
| Rimed stellar (R1d) | 48 | 0.70–5.3 mm | 2.0–539 µg | – | 13. $m\,/(\mathrm{\mu g}) = 5.34 \cdot (D/\mathrm{mm})^{2.58}$ | 0.85 | H87 |
| Unrimed dendrites | – | – | – | – | 14. $m\,/(\mathrm{\mu g}) = 15.5 \cdot (D/\mathrm{mm})^{1.91}$ | – | [E] |
| Rimed dendrites | – | – | – | – | 15. $m\,/(\mathrm{\mu g}) = 32.7 \cdot (D/\mathrm{mm})^{1.78}$ | – | E17 |
| Stellar with end plates (P2a) | 97 | 1.4–7 mm | – | – | 16. $m\,/(\mathrm{\mu g}) = 6.75 \cdot (D/\mathrm{mm})^{2.09}$ | $r = 0.76$ | K89 |
| Rimed stellar (R1d) | 43 | 1.6–5.8 mm | – | – | 17. $m\,/(\mathrm{\mu g}) = 9.18 \cdot (D/\mathrm{mm})^{1.76}$ | $r = 0.68$ | K89 |
| **Shape group *(12) Graupel*** | 37 | 0.25–1.2 mm | 1.31–68 µg | 0.26–1.0 m s$^{-1}$ | 18. $m\,/(\mathrm{\mu g}) = 53.9 \cdot (D/\mathrm{mm})^{2.74}$ | 0.98 | VM21 |
| | 37 | 0.25–1.2 mm | 1.31–68 µg | 0.26–1.0 m s$^{-1}$ | 19. $v\,/(\mathrm{m\,s^{-1}}) = 0.24 \cdot (m/\mathrm{\mu g})^{0.39}$ | 0.94 | VM21 |
| Lump graupel (R4b) | 35 | 0.50–2.0 mm | – | – | 20. $m\,/(\mathrm{\mu g}) = 42.0 \cdot (D/\mathrm{mm})^{3.00}$ | $r = 0.98$ | L74 |
| | 35 | 0.50–2.0 mm | – | – | 21. $v\,/(\mathrm{m\,s^{-1}}) = 0.46 \cdot (m/\mathrm{\mu g})^{0.15}$ | $r = 0.53$ | L74 |
| Lump graupel (R4b) | 58 | 0.50–3.0 mm | – | – | 22. $m\,/(\mathrm{\mu g}) = 78.0 \cdot (D/\mathrm{mm})^{2.80}$ | $r = 0.93$ | L74 |
| | 58 | 0.50–3.0 mm | – | – | 23. $v\,/(\mathrm{m\,s^{-1}}) = 0.46 \cdot (m/\mathrm{\mu g})^{0.24}$ | $r = 0.84$ | L74 |
| Lump graupel (R4b) | 17 | 0.50–1.0 mm | – | – | 24. $m\,/(\mathrm{\mu g}) = 140 \cdot (D/\mathrm{mm})^{2.70}$ | $r = 0.98$ | L74 |
| | 17 | 0.50–1.0 mm | – | – | 25. $v\,/(\mathrm{m\,s^{-1}}) = 0.79 \cdot (m/\mathrm{\mu g})^{0.12}$ | $r = 0.52$ | L74 |
| Lump graupel (R4b) | 116 | 0.40–9.0 mm | 14–68,000 µg | – | 26. $m\,/(\mathrm{\mu g}) = 85.0 \cdot (D/\mathrm{mm})^{3.10}$ | 0.89 | H87 |
| Lump graupel (R4b) | – | – | – | – | 27. $m\,/(\mathrm{\mu g}) = 53.7 \cdot (D/\mathrm{mm})^{2.16}$ | – | E17 |
| **Shape group *(15) Spherical*** | 41 | 0.06–0.4 mm | 0.16–39 µg | 0.09–1.6 m s$^{-1}$ | 28. $m\,/(\mathrm{\mu g}) = 260 \cdot (D/\mathrm{mm})^{2.84}$ | 0.88 | VM21 |
| | 41 | 0.06–0.4 mm | 0.16–39 µg | 0.09–1.6 m s$^{-1}$ | 29. $v\,/(\mathrm{m\,s^{-1}}) = 0.30 \cdot (m/\mathrm{\mu g})^{0.51}$ | 0.99 | VM21 |

## Appendix A: Mass relationships for the shape groups

Figures A1–A3 shows the $m$ vs $D_\mathrm{max}$, $m$ vs $A$, and $v$ vs $m$ relationships for all the 15 shape groups fitted to binned data. These relationships correspond to power laws given by Eq. 8–Eq. 10.

## Appendix B: Mass derivation using power laws

The particle mass relationships are derived analytically from a relationship between the Reynolds and Best numbers, in addition to $A$ vs $D_\mathrm{max}$, $v$ vs $D_\mathrm{max}$, and $v$ vs $A$ power laws given by Eq. 11a–Eq. 13a. Section 3 has briefly presented this approach of deriving the particle mass analytically. The $m$ vs $D_\mathrm{max}$, $m$ vs $A$, and $v$ vs $m$ relationships given by this approach are equivalent to fitting to individual data. Indeed we get identical results in the $\tilde{a}_D$, $\tilde{b}_D$, $\tilde{a}_A$, $\tilde{b}_D$, $a_m$, $b_m$ parameters if using $X^*$ vs Re as power law

$$X^*(\mathrm{Re}) = \gamma \cdot \mathrm{Re}^\delta, \tag{B1}$$

where $\gamma$ and $\delta$ are the parameters in the power law. We determine these parameters by fitting the power law to Eq. 5 over ranges of Re corresponding to each shape group. For this, we first calculate Re for all particles in a shape group and determine $X^*$ using Eq. 5 for this set of Re values. Then, we do a linear fit to the logarithm of $X^*$ vs logarithm of Re. Consequently, for each shape group, we get one set of $\gamma$ and $\delta$.

We express Re as a power law in $D_\mathrm{max}$ using Eq. 2 and replacing $v$ with the power law given by Eq. 12a

$$\mathrm{Re}(v, D_\mathrm{max}) = \frac{v(D_\mathrm{max}) \cdot D_\mathrm{max} \cdot \rho_\mathrm{a}}{\eta} = \frac{\left(a_D \cdot \left(\frac{D_\mathrm{max}}{1\,\mathrm{mm}}\right)^{b_D}\right) \cdot \frac{D_\mathrm{max}}{1\,\mathrm{mm}} \cdot 1\,\mathrm{mm} \cdot \rho_\mathrm{a}}{\eta} =$$
$$= \left(\frac{a_D \cdot \rho_\mathrm{a} \cdot 1\,\mathrm{mm}}{\eta}\right) \cdot \left(\frac{D_\mathrm{max}}{1\,\mathrm{mm}}\right)^{b_D+1}. \tag{B2}$$

Now we can determine the particle mass $m$ using Eq. 7 and express it as a function of particle size $D_\mathrm{max}$, area $A$, or fall speed $v$. Consequently, the mass relationship as a function of particle size $D_\mathrm{max}$ given by Eq. 8 can be derived as follows (using Eq. B1, Eq. B2, Eq. 11a, and the area ratio $A_\mathrm{r} = \frac{A}{\frac{\pi}{4} \cdot D_\mathrm{max}^2}$)

$$m(D_\mathrm{max}) = \frac{\pi \cdot \eta^2 \cdot X^* \cdot A_\mathrm{r}^{1/2}}{8 \cdot g \cdot \rho_\mathrm{a}} = \frac{\pi \cdot \eta^2 \cdot \gamma \cdot \mathrm{Re}^\delta \cdot \left(\frac{A(D_\mathrm{max})}{\frac{\pi}{4} \cdot D_\mathrm{max}^2}\right)^{1/2}}{8 \cdot g \cdot \rho_\mathrm{a}}$$

$$= \frac{\pi \cdot \eta^2 \cdot \gamma \cdot \left(\frac{a_D \cdot \rho_\mathrm{a} \cdot 1\,\mathrm{mm}}{\eta}\right)^\delta \cdot \left(\frac{D_\mathrm{max}}{1\,\mathrm{mm}}\right)^{(b_D+1)\cdot\delta}}{8 \cdot g \cdot \rho_\mathrm{a}} \cdot \left(\frac{a \cdot \left(\frac{D_\mathrm{max}}{1\,\mathrm{mm}}\right)^b}{\frac{\pi}{4} \cdot \left(\frac{D_\mathrm{max}}{1\,\mathrm{mm}} \cdot 1\,\mathrm{mm}\right)^2}\right)^{1/2}$$

$$= \frac{\pi^{1/2} \cdot \eta^2 \cdot \gamma}{4 \cdot g \cdot \rho_\mathrm{a}} \cdot \left(\frac{a}{1\,\mathrm{mm}^2}\right)^{1/2} \cdot \left(\frac{a_D \cdot \rho_\mathrm{a} \cdot 1\,\mathrm{mm}}{\eta}\right)^\delta \cdot \left(\frac{D_\mathrm{max}}{1\,\mathrm{mm}}\right)^{b_D \cdot \delta + \delta + \frac{1}{2} \cdot b - 1} = \tilde{a}_D \cdot \left(\frac{D_\mathrm{max}}{1\,\mathrm{mm}}\right)^{\tilde{b}_D}. \tag{B3}$$

The mass relationship as a function of $A$ given by Eq. 9 can be derived as follows (using Eq. B1, Eq. 11b, and Eq. 13a, and expressing Re as a power law in $A$)

$$m(A) = \frac{\pi \cdot \eta^2 \cdot X^* \cdot A_{\rm r}^{1/2}}{8 \cdot g \cdot \rho_{\rm a}} = \frac{\pi \cdot \eta^2 \cdot \gamma \cdot \left( \frac{v(A) \cdot D_{\max}(A) \cdot \rho_{\rm a}}{\eta} \right)^\delta}{8 \cdot g \cdot \rho_{\rm a}} \cdot \left( \frac{\frac{A}{1 \, {\rm mm}^2} \cdot 1 \, {\rm mm}^2}{\frac{\pi}{4} \cdot \left[ a' \cdot \left( \frac{A}{1 \, {\rm mm}^2} \right)^{b'} \right]^2} \right)^{1/2}$$

$$= \frac{\pi^{1/2} \cdot \eta^2 \cdot \gamma \cdot \left( \frac{a' \cdot \left( \frac{A}{1 \, {\rm mm}^2} \right)^{b'} \cdot \left[ a' \cdot \left( \frac{A}{1 \, {\rm mm}^2} \right)^{b'} \right] \cdot \rho_{\rm a}}{\eta} \right)^\delta}{4 \cdot g \cdot \rho_{\rm a}} \cdot \left( \frac{\frac{A}{1 \, {\rm mm}^2} \cdot 1 \, {\rm mm}^2}{\left[ a' \cdot \left( \frac{A}{1 \, {\rm mm}^2} \right)^{b'} \right]^2} \right)^{1/2}$$

$$= \frac{\pi^{1/2} \cdot \eta^2 \cdot \gamma}{4 \cdot g \cdot \rho_{\rm a}} \cdot \frac{1 \, {\rm mm}}{a'} \cdot \left( \frac{a_A \cdot a' \cdot \rho_{\rm a}}{\eta} \right)^\delta \cdot \left( \frac{A}{1 \, {\rm mm}^2} \right)^{b_A \cdot \delta + b' \cdot \delta + 1/2 - b'} = \tilde{a}_A \cdot \left( \frac{A}{1 \, {\rm mm}^2} \right)^{\tilde{b}_A}. \tag{B4}$$

The mass relationship as a function of $v$ given by Eq. 10 can be derived as follows (using Eq. B1, Eq. 12b, and Eq. 13b, and expressing Re as a power law in $v$)

$$m(v) = \frac{\pi \cdot \eta^2 \cdot X^* \cdot A_{\rm r}^{1/2}}{8 \cdot g \cdot \rho_{\rm a}} = \frac{\pi \cdot \eta^2 \cdot \gamma \cdot \left( \frac{v \cdot D_{\max}(v) \cdot \rho_{\rm a}}{\eta} \right)^\delta}{8 \cdot g \cdot \rho_{\rm a}} \cdot \left( \frac{A(v)}{\frac{\pi}{4} \cdot D_{\max}(v)^2} \right)^{1/2}$$

$$= \frac{\pi \cdot \eta^2 \cdot \gamma}{8 \cdot g \cdot \rho_{\rm a}} \cdot \left( \frac{v}{1 \, {\rm m \, s}^{-1}} \cdot 1 \, {\rm m \, s}^{-1} \cdot a_D' \cdot \left( \frac{v}{1 \, {\rm m \, s}^{-1}} \right)^{b_D'} \cdot \frac{\rho_{\rm a}}{\eta} \right)^\delta \cdot \left( \frac{a_A' \cdot \left( \frac{v}{1 \, {\rm m \, s}^{-1}} \right)^{b_A'}}{\frac{\pi}{4} \cdot \left( a_D' \cdot \left( \frac{v}{1 \, {\rm m \, s}^{-1}} \right)^{b_D'} \right)^2} \right)^{1/2}$$

$$= \frac{\pi^{1/2} \cdot \eta^2 \cdot \gamma \cdot a_A'^{1/2}}{4 \cdot g \cdot \rho_{\rm a} \cdot a_D'} \cdot \left( a_D' \cdot 1 \, {\rm m \, s}^{-1} \cdot \frac{\rho_{\rm a}}{\eta} \right)^\delta \cdot \left( \frac{v}{1 \, {\rm m \, s}^{-1}} \right)^{\delta + b_D' \cdot \delta + b_A'/2 - b_D'}. \tag{B5}$$

From Eq. B5, we can determine $v(m)$ as follows

$$\left( \frac{v}{1 \, {\rm m \, s}^{-1}} \right)^{b_D' \cdot \left( \delta - \frac{1}{2} \right) + \frac{1}{2} \cdot b_A'} = \frac{m}{1 \, \mu{\rm g}} \cdot 1 \, \mu{\rm g} \cdot \frac{4 \cdot g \cdot \rho_{\rm a} \cdot a_D'}{\pi^{1/2} \cdot \eta^2 \cdot \gamma \cdot a_A'^{1/2}} \cdot \left( a_D' \cdot 1 \, {\rm m \, s}^{-1} \cdot \frac{\rho_{\rm a}}{\eta} \right)^{-\delta}$$

$$\Rightarrow v(m) = 1 \, {\rm m \, s}^{-1} \cdot \left[ \frac{4 \cdot g \cdot \rho_{\rm a} \cdot a_D' \cdot 1 \, \mu{\rm g}}{\pi^{1/2} \cdot \eta^2 \cdot \gamma \cdot a_A'^{1/2}} \cdot \left( a_D' \cdot 1 \, {\rm m \, s}^{-1} \cdot \frac{\rho_{\rm a}}{\eta} \right)^{-\delta} \right]^{\frac{1}{b_D' \cdot \left( \delta - \frac{1}{2} \right) + \frac{1}{2} \cdot b_A'}} \cdot \left( \frac{m}{1 \, \mu{\rm g}} \right)^{\frac{1}{b_D' \cdot \left( \delta - \frac{1}{2} \right) + \frac{1}{2} \cdot b_A'}}$$

$$= a_m \cdot \left( \frac{m}{1 \, \mu{\rm g}} \right)^{b_m}. \tag{B6}$$

## Appendix C: Closure study—Reynolds and Best numbers for simple thick columns

Selecting a simple shape with area ratio noticeably below 1, we can test if the modified Best number approach by Heymsfield and Westbrook (2010) yields better results than using Best numbers and the approach by Böhm (1989) (see Sect. 3.1 for details about these approaches). If Reynolds and Best numbers (Re and $X$, respectively) can be determined independently, i.e., without using any $X$–Re relationship, then they can be compared with the $X$–Re relationship by Böhm (1989) given by Eq. 5. Thus, it represents a closure study that can confirm the $X$–Re relationship, which this and other studies rely on. The following explains

the method and results when applied to simple columns, a small subset of our data.

For a simple-geometry shape we can calculate the particle mass from the geometrical dimensions and, thus, determine both $X$ and Re independently. Then $X$ vs Re, or alternatively $X^*$ vs Re can be compared to the empirical relationship given by Eq. 4 or Eq. 5. Needles or columns would be suitable shapes as they have low area ratios and a simple geometry. Looking at particles in the shape group *(1) Needles* reveals that it contains many bundles of needles and only few pristine needles. Shape group *(3) Thick columns*, on the other hand, contains many simple columns. Therefore, we have selected 75 columns from shape group *(3)* for this comparison study. Figure C1 shows examples of the selected columns.

Most columns fall horizontally so that width and length can be easily determined from the top-view images. We estimate that the length may be underestimated on the order of up 15% due to deviations from alignment of the column axis in the image plane. On the other hand, the geometrically determined mass, $m_{\mathrm{geom}}$, may be overestimated for part of the columns that show signs of cavities or hollowing of faces (see Fig. C1).

For columns, $D_{\mathrm{max}}$, which is similar to the column's length, is not a suitable representative size parameter to determine Re, as we have discussed in Sect. 4.2.1 and Vázquez-Martín et al. (2021). A characteristic length $L^* = A_{\mathrm{t}}/P$, where $A_{\mathrm{t}}$ is the total surface area and $P$ the perimeter of the particle projected to the flow (see Eq. 13-81 in Pruppacher and Klett, 2010), can be used instead. For columns, $L^*$ can be determined from width and length (Jayaweera, 1971) and is more closely related to the width. Now, Re can be determined from measured fall speed and $L^*$. The Best number, according to Eq. 3, can be determined from measured cross-sectional area $A$ and $D_{\mathrm{max}}$. Note that here, $D_{\mathrm{max}}$ represents the same size parameter best suited to calculate Re as in Eq. 2. Thus, not only Re should be determined from $L^*$ (instead of $D_{\mathrm{max}}$), but also $X$. Then, $X$ can be determined from measured $A$ in addition to calculated $m_{\mathrm{geom}}$ and $L^*$.

Consequently, $X$ vs Re can be plotted and compared to the $X$–Re relationship (Eq. 5). Figure C2 shows $X$ vs Re determined either using $D_{\mathrm{max}}$ or $L^*$. The points related to $D_{\mathrm{max}}$ (blue triangles) do not match well the empirical relationship $X$–Re (with $\delta_0 = 5.83$ and $C_0 = 0.6$) by Böhm (1989) based on a theoretical treatment by Abraham (1970). This confirms that, as argued above, $D_{\mathrm{max}}$ is not suitable to determine Re or $X$ for this shape. The points $X$ vs Re determined using $L^*$, on the other hand, are much closer to the empirical relationship. Thus, this closure experiment comparing independently determined Re and $X$ to the $X$–Re relationship demonstrates the superiority of characteristic length $L^*$ over $D_{\mathrm{max}}$ as a particle size parameter when dealing with particle mass $m$.

The points $X$ vs Re can be transformed into $X^*$ vs Re where $X^* = X \cdot A_{\mathrm{r}}^{1/2}$ is the modified Best number suggested by Heymsfield and Westbrook (2010). The resulting points (using $L^*$) are also shown in Fig. C2 (green 'x') and provide an even better closure to the empirical $X$–Re relationship. Heymsfield and Westbrook (2010) used $D_{\mathrm{max}}$ and not characteristic length $L^*$ (they focused on shapes with open geometries for which characteristic length is difficult to determine). The closure for our columns using $X^*$ and $D_{\mathrm{max}}$ (cyan crosses in Fig. C2) is not as good as using $X^*$ and $L^*$, but still better than using the unmodified Best number $X$ and $D_{\mathrm{max}}$. Thus, for columns we can conclude that the modified Best number represents an improvement over the Best number. In addition, the superiority of characteristic length $L^*$ over $D_{\mathrm{max}}$ for columns is given also when working with the modified Best number $X^*$.

In addition to the empirical relationship $X$–Re by Böhm (1989), also the relationship by Heymsfield and Westbrook (2010) ($\delta_0 = 8.0$ and $C_0 = 0.35$) for their the modified Best number approach, used in our study, is shown in Fig. C2. The two

lines are relatively close to each other. Thus, the above conclusions of superiority of characteristic length $L^*$ over $D_{\max}$ and improvement when using modified Best number rather than Best number remain valid regardless of which relationship is used as comparison.

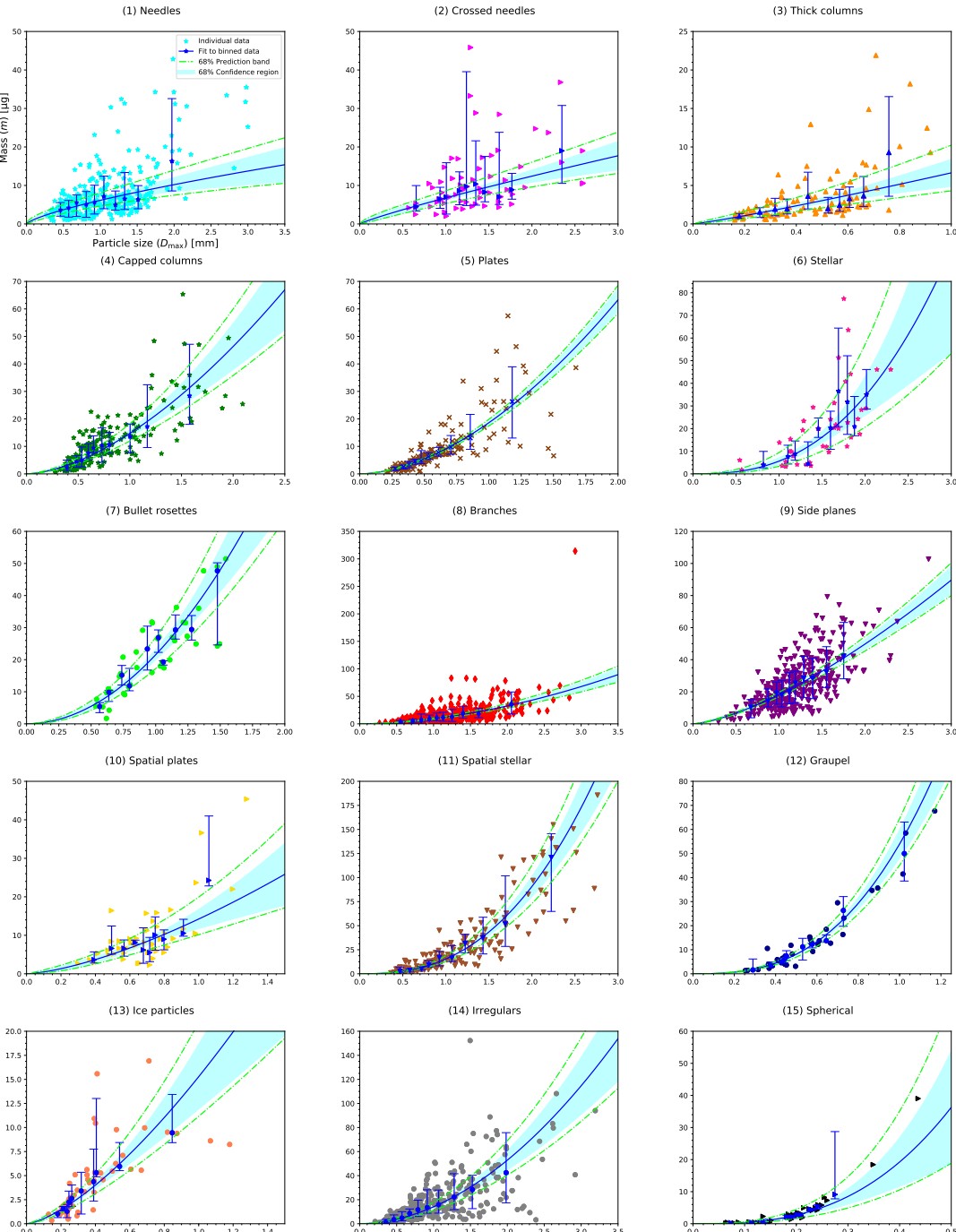

**Figure A1.** Mass vs particle size ($m$ vs $D_{max}$) relationships given by Eq. 8–Eq. 10 for all the shape groups are shown. Individual data (coloured symbols) and binned data (blue symbols with error bars) are displayed. Median values in the respective bins represent the binned data. The total length of the error bars represents the spread in mass data, which is given by the difference between the $16^{th}$ and $84^{th}$ percentiles. The relationships fitted to binned data are shown. The 68% prediction band and the 68% confidence region for the fits are also shown. The same data are shown in Table 1.

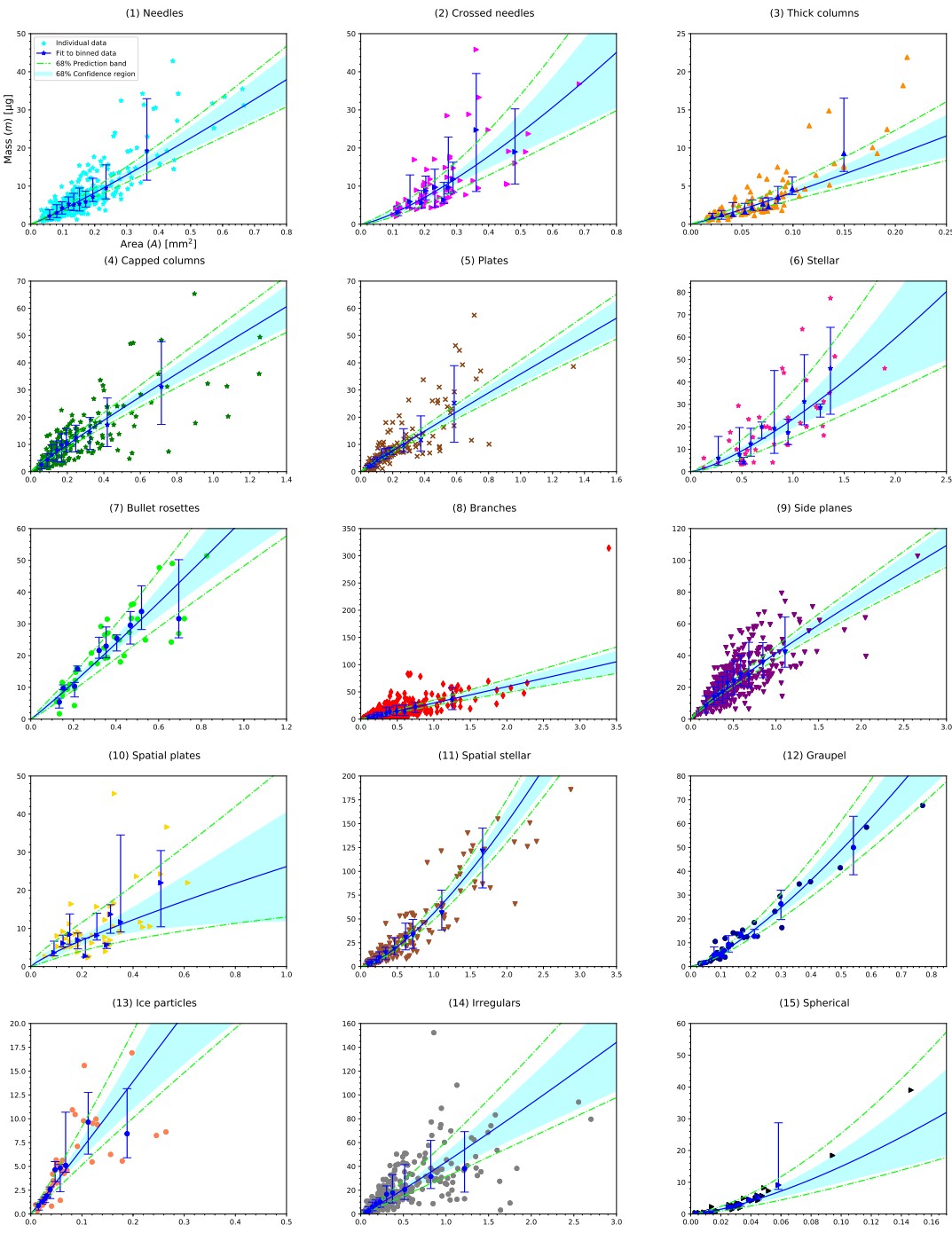

**Figure A2.** Same as Figure A1, but mass vs cross-sectional area ($m$ vs $A$) relationships given by Eq. 8–Eq. 10 are shown here.

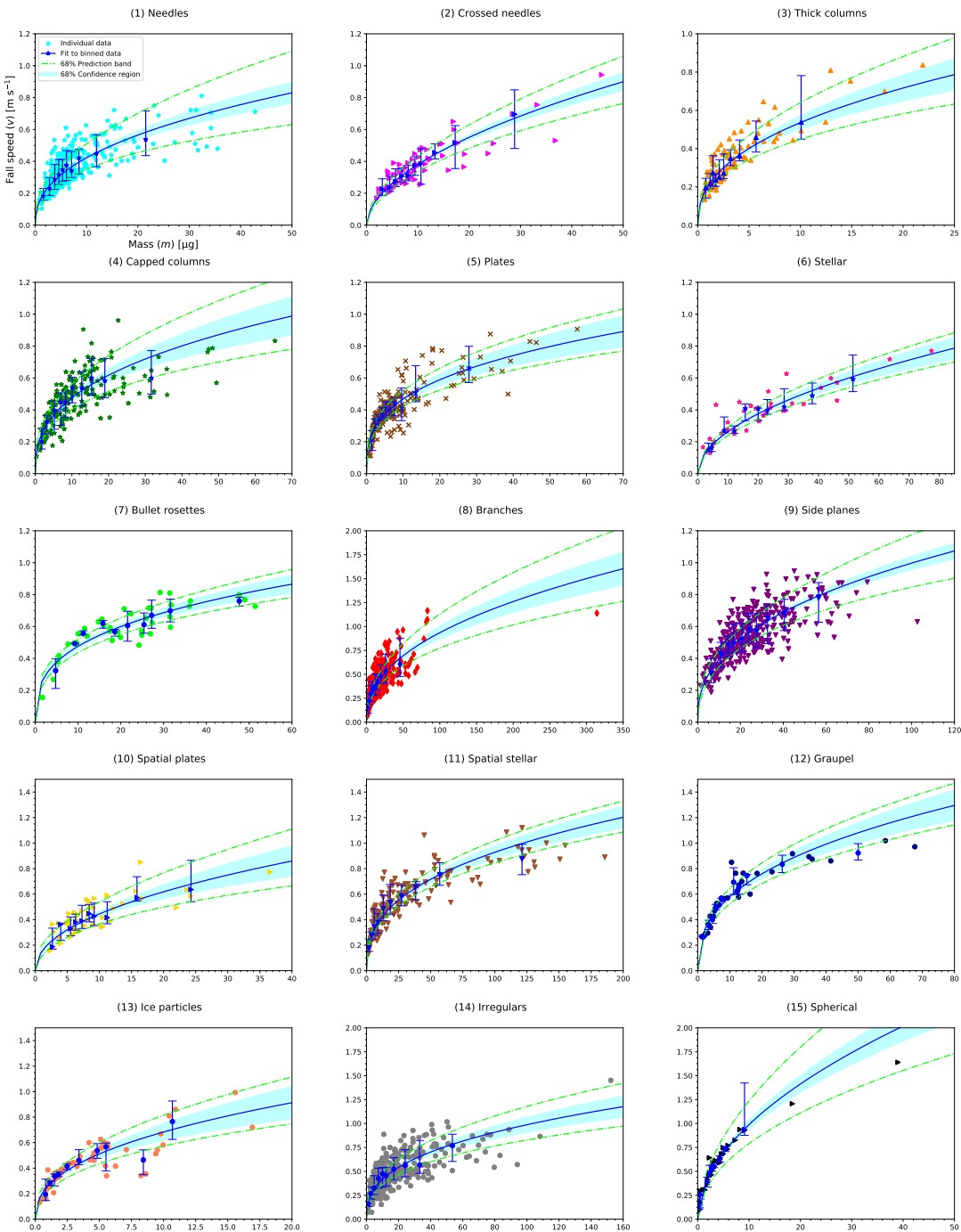

**Figure A3.** Same as Figures A1–A1, but fall speed vs mass ($v$ vs $m$) relationships given by Eq. 8–Eq. 10 are shown here.

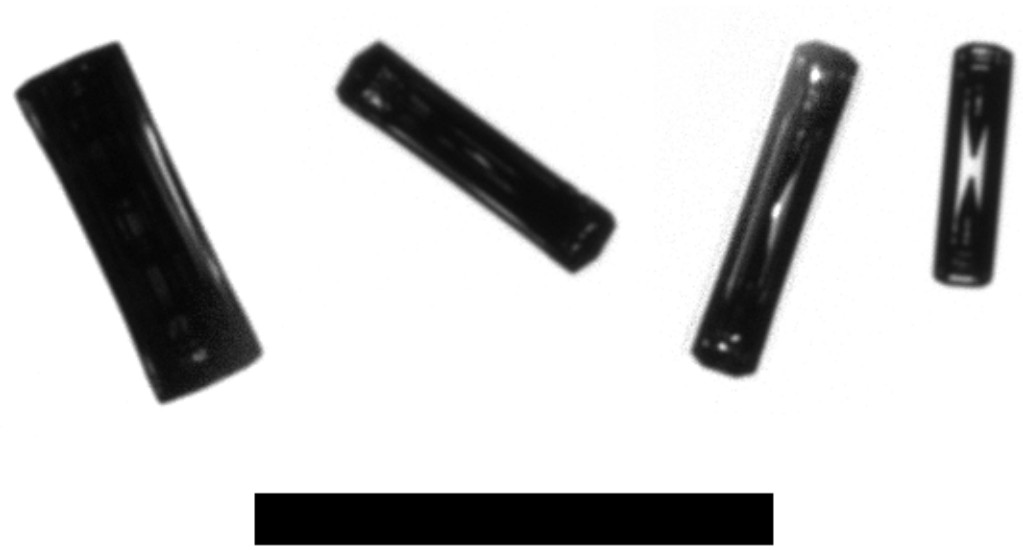

**Figure C1.** Examples of simple thick columns selected from shape group *(3) Thick columns*. The black rectangle shown as size reference corresponds to 1 mm × 100 μm.

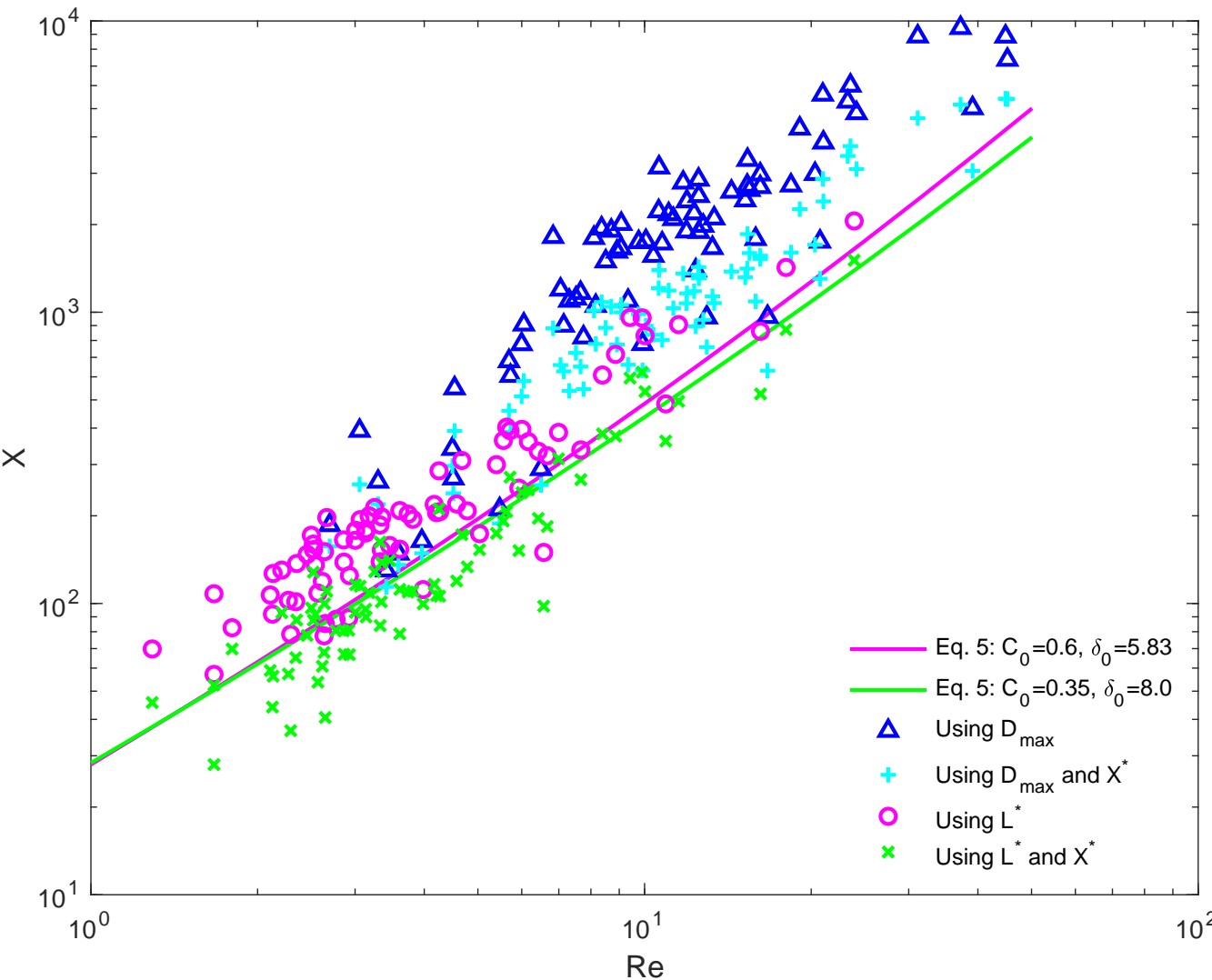

**Figure C2.** $X$ vs $Re$ and $X^*$ vs $Re$ for simple thick columns selected from shape group *(3) Thick columns*. $X$ and $Re$ are determined either using $D_{max}$ or $L^*$. The points $X$ vs $Re$ using $L^*$ are much closer to the empirical relationship (Eq. 5) than the points using $D_{max}$. Using the modified Best number $X^*$ instead of the Best number $X$ leads to a better agreement with Eq. 5. For comparison, the empirical relationship given by Eq. 5 is shown with parameters from Böhm (1989) and Heymsfield and Westbrook (2010), respectively. They are, however, very similar.

## Appendix: Nomenclature

### Latin Letters

| | |
|---|---|
| $A$ | cross-sectional area |
| $a$ | coefficient in the $A$ vs $D_{\max}$ relationship |
| $a'$ | coefficient in the $D_{\max}$ vs $A$ relationship |
| $a_A$ | coefficent in the $v$ vs $A$ relationship |
| $a'_A$ | coefficient in the $A$ vs $v$ relationship |
| $\tilde{a}_A$ | coefficient in the $m$ vs $A$ relationship |
| $a_D$ | coefficient in the $v$ vs $D_{\max}$ relationship |
| $a'_D$ | coefficient in the $D_{\max}$ vs $v$ relationship |
| $\tilde{a}_D$ | coefficient in the $m$ vs $D_{\max}$ relationship |
| $a_m$ | coefficient in the $v$ vs $m$ relationship |
| $A_{\mathrm{t}}$ | total surface area |
| $b$ | exponent in the $A$ vs $D_{\max}$ relationship |
| $b'$ | exponent in the $D_{\max}$ vs $A$ relationship |
| $b_A$ | exponent in the $v$ vs $A$ relationship |
| $b'_A$ | exponent in the $A$ vs $v$ relationship |
| $\tilde{b}_A$ | exponent in the $m$ vs $A$ relationship |
| $b_D$ | exponent in the $v$ vs $D_{\max}$ relationship |
| $b'_D$ | exponent in the $D_{\max}$ vs $v$ relationship |
| $\tilde{b}_D$ | exponent in the $m$ vs $D_{\max}$ relationship |
| $b_m$ | exponent in the $v$ vs $m$ relationship |
| $C_0$ | unit-less constant in $\mathrm{Re}$ vs $X$ relationship |
| $C_{\mathrm{D}}$ | drag coefficient |

$g$      acceleration of gravity

$L^*$      characteristic length

$m$      particle mass

$P$      perimeter of projected particle image

$R_A^2$      correlation coefficient in the $m$ vs $A$ relationship

$R_D^2$      correlation coefficient in the $m$ vs $D_{\mathrm{max}}$ relationship

$R_m^2$      correlation coefficient in the $v$ vs $m$ relationship

$\mathrm{Re}$      Reynolds number

$v$      fall speed

$X$      Best number

**Greek Letters**

$\delta$      exponent in the $X^*$ vs $\mathrm{Re}$ relationship

$\delta_0$      unit-less constant in $\mathrm{Re}$ vs $X$ relationship

$\eta$      dynamic viscosity of air

$\gamma$      coefficient in the $X^*$ vs $\mathrm{Re}$ relationship

$\rho_{\mathrm{a}}$      air density

$\rho_{\mathrm{w}}$      density of liquid water

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
