# Peer review of "Mass of different snow crystal shapes derived from fall speed measurements"

_Atmospheric Chemistry and Physics, 2021_

## Referee Comment (RC1)

Referee Report for ACP
ACPD article title: Mass of different snow crystal shapes derived from fall speed measurements
Author(s): Sandra Vázquez-Martín et al.
MS No.: acp-2021-203
MS type: Research article

**General Comments:**

This paper is the first to demonstrate the calculation of ice particle mass from measurements of ice particle maximum dimension, projected area and fall velocity, and in doing so, it represents a test of hydrodynamic flow theory. However, as argued below, the method appears successful for graupel and quasi-spherical ice particles, but less successful for planar ice crystals (e.g., stellars or dendrites) and definitely not successful for needles and columnar ice crystals. Similar findings were reported by Heymsfield and Westbrook (2010, JAS; henceforth HW2010), where the fall speed treatment of Mitchell (1996, JAS; henceforth M96) worked well for ice crystals having aspect ratios closer to unity (e.g., graupel, short columns, thick plates) but not well for stellars and needles having more extreme aspect ratios. Therefore, two new approaches are offered for modifying the methodology described in this study.

One approach is to define the Reynolds number Re in terms of a characteristic length $L^*$, rather than maximum dimension $D_{max}$, where $Re_{L^*} = V L^*/\nu$, where V = terminal fall speed and $\nu$ = kinematic viscosity of air = $\eta/\rho_a$ where $\rho_a$ = air density and $\eta$ = dynamic viscosity of air. This $L^*$ was found to describe the vapor mass and heat transfer from a ventilated ice crystal well and hence captures the flow effect on ice crystal growth. $L^*$ is defined as

$$L^* = A/P \tag{1}$$

where A = total surface area of an ice particle and P = ice particle perimeter projected to the flow. See Pruppacher and Klett (1997), *Microphysics of Clouds and Precipitation*, Kluwer Academic Publishers, p. 552, for more details.

Fortunately, Jayaweera (1971, JAS) has formulas describing $L^*$ for planar and columnar ice crystals. For planar ice crystals, $L^* = (d/2) (1 + 2e)$ and

$$Re_{L^*} = 0.5(1 + 2e) Re_d \tag{2}$$

where subscript d on Reynolds number Re indicates that Re is evaluated using the diameter d of a circle having the same area as the basal face of the ice crystal. For hexagonal plates, d = 0.910 D, where D = maximum dimension of the basal face. Moreover, e refers to the ratio of minor to major axis. Reasonable estimates for e can be obtained from $D_{max}$ and Auer and Veal (1970, JAS). For columnar ice crystals, $L^* = (\pi/4)d [1 + (1/(1+e))]$ and

$$Re_{L^*} = (\pi/4) [1 + (1/(1+e))] Re_d . \tag{3}$$

Notice here that $Re_{L*}$ depends on the "diameter" or thickness of a column and not its maximum dimension $D_{max}$ (as was used in this ACPD paper). This indicates V is most related to d. By substituting $Re_{Dmax}$ used in this ACPD paper with $Re_{L*}$ in Eqn. 5 to calculate Best number "X", and substituting $D_{max}$ with L* in Eqn. 6 of this paper (while using this new calculation for X), the m-$D_{max}$ power laws found here for columnar and planar ice crystals may be improved, having greater consistency with the body of theoretical and empirical knowledge (discussed below under Major Comments).

The second approach is to calculate the "modified Best number" X* from Eqn. 5 of this ACPD paper (as described in HW2010) by redefining the constants used to calculate X*, where $C_0 = 0.35$ and $\delta_0 = 8.0$. However, in this case $Re = Re_{Dmax}$ (as originally used in this paper) since this maintains consistency with HW2010. Then mass m is calculated by inverting the HW2010 definition of X*:

$$m = \pi \, \eta^2 \, X^* \, A_r^{1/2}/( 8 \, g \, \rho_a ) \tag{4}$$

where m = ice particle mass, g = gravity constant and $A_r$ = area of ice crystal normal to the flow divided by area of circle having same maximum dimension (referred to as the area ratio).

However, if the "thick columns" shape category in this study corresponds to short, thick columns, the M96 fall speed scheme should work fine for this shape category, and this second approach may not address the problem for this shape.

It is possible that neither of these alternative approaches will render improved results, but it seems worth a try. If there is no improvement, the limitations described below will need to be mentioned in the paper.

**Major Comments:**

1. Lines 149-150: Please indicate which relationships in Tables 1 and 2 are based on the approach described in Sect. 3.3 vs. the approach given in Sect. 3.2. Since this approach involves empirical relationships already having considerable uncertainty, m(D), m(A) and v(m) from this approach may have greater uncertainty than the previous approach (Sect. 3.2) due to the propagation of uncertainties in the empirical expressions used here. This knowledge may be helpful for interpreting the results in Tables 1 and 2.

2. Lines 185-188 on stellar ice crystals: This same argument is expressed more quantitatively in Mitchell et al. (1990, Sect. 4a), where hexagonal crystal volume V is approximated by using a circular basal face so that aspect ratio k = c/a = c/r, where r = radius of this circle having the same area as the basal face. Moreover, c and a are the semi-axes corresponding to the prism and basal faces of an ice crystal. Thus, $V = \pi \, r^2 \, k \, r = \pi \, k \, r^3$ for this approximation, and for constant density and constant k, the m-D relationship has a power of 3.

However, constant k is an invalid assumption for a stellar ice crystal that only forms between ~ -14 and -16 °C.  As described in Chen and Lamb (1994, JAS), k is rarely constant (definitely not at these temperatures), and depends on the ratio of condensation coefficients for the basal and prism crystal faces.  This is referred to as the inherent growth ratio (IGR), and IGR is related to m-D power laws in Sect. 7 of their paper.  I strongly recommend that the authors read this paper and then revise this commentary accordingly.  This information is also in the cloud physics text book by Lamb and Verlinde, Physics and Chemistry of Clouds (2011, Cambridge Univ. Press).  In addition, Jerry Harrington's group at Penn. State Univ. has greatly extended this work through several publications (e.g., Harrington et al., 2013, JAS, "A method for adaptive habit prediction in bulk microphysical models. Part I: Theoretical development).

Physical intuition also informs us that slope $b_D$ is too high since $b_D$ is a measure of the increase in mass with respect to size.   A large (i.e., steep) slope indicates a relatively large mass increase per unit size increment, but this is not true for stellar or dendrite ice crystals since their ice density decreases with increasing size.

3. Lines 190-196:  These shapes are all columnar, which may be a clue to the problem here. Reynolds Number Re is expressed by (2) in terms of $D_{max}$, but due to hydrodynamical considerations, some argue that $D_{max}$ should be replaced by a "characteristic length" L* defined in Pruppacher and Klett (1997) as: L* = A/P, where A = total ice particle surface area and P = particle perimeter normal to the flow.  For needles, changes in A and P will be roughly proportional, with L* changing much less than $D_{max}$.

As shown by Jayaweera (1971, JAS), L* is strongly related to the column radius (or basal face semi-axis) and weakly related to $D_{max}$, indicating the formulation of Eqns. (2) and (5) in terms of $D_{max}$ are flawed based on $Re_{L*}$.  This is indeed the case for hexagonal columns.

4. Lines 196-199:  Note that $b_D$ < 1.0 also for thick columns (C1e; group 3), with $b_D$ = 0.81. Compare this with Tables 1 & 3 in Mitchell et al. (1990, JAM)), where C1e $b_D$ = 2.6 is found to be consistent with other studies based on dimensional-density relationships. Moreover, $b_D$ = 2.6 is very consistent with the theoretical prediction of Chen and Lamb for C1e $b_D$ (see their Fig. 12). This is strong evidence that the C1e $b_D$ in this current study suffers from some limitation.  Please expose this issue for the readers.

5. Section 4.4.1 on plates:  Chen and Lamb (1992) provide theoretical limits for columnar and planar ice crystals regarding $b_D$, where $b_D$ for hexagonal plates lies between 2.0 and 3.0.  In this current study for plates, $b_D$ = 1.72, indicating its value should be treated with caution; please make readers aware of this.  Since side planes grow diffusionally through a different mechanism (Furakawa, 1982, J. Meteor. Soc. Japan), it is not clear whether these limits apply to side planes.  Please mention this.

6. Figure 3 caption, last sentence: The text indicates that this should be valid for all ice particles and not just spheres; please make this clear.

7. Lines 290-291: Can it be said that Re-X represents the fall speed upper limit?

8. Summary and conclusions; 1st bullet: Will the fact that m is derived from v produce a co-variance that contributes to the stronger correlation between v and m? If so, please mention this wherever it is most appropriate.

9. Summary and conclusions; 2nd bullet: Will not the power-law approximations have greater uncertainty than the relationships based on Eq. 5? Please address this concern wherever it is most appropriate.

10. Summary and conclusions; last sentence of 3rd bullet: But we know this is not true based on Chen and Lamb (1994, JAS) and other m-D measurements for stellar crystals. Please remove this last sentence.

11. Line 328-330: I don't see this relationship plotted (spherical ice having a density of 0.12 g cm$^{-3}$). If it is not shown, then please add in parentheses, "not shown".

**Minor Comments:**

1. Line 150: Apparent typo; Sect. 3.1 => 3.2?

2. Lines 248-9: It appears that Ma has not been defined.

3. Line 329: The second comma is not needed.

Best wishes for a successful study,
David Mitchell

---

## Author Comment (AC2)

**Referee #2**
* * *
Dear referee,

We thank you for your constructive feedback and appreciate the time you spent to read and evaluate our work. Please see below our response to your comments.

*We are reporting our response (in blue italics) directly following each point that you have raised.* Then, we are suggesting changes to the manuscript (still in blue).

- **General comments:**

I have read the manuscript "Mass of different snow crystal shapes derived from fall speed measurements" by Vázquez-Martín, Kuhn, and Eliasson and I find it an interesting manuscript. The authors use a dataset of measurements of 2461 ice crystals including fall speed, area, and maximum dimension. From these data they derive particle mass using standard Reynolds – Best number approaches. The dataset is then broken down into 15 different particle habit classifications and relationships between maximum dimension and the other parameters are presented and compared to previous literature.

While it is a nice study that merits being published, there are a few shortcomings that I feel should be addressed before full acceptance.

1) The project is highly dependent on the Re – X number relationship for relating mass and terminal velocity. Traditionally, this is used in the other direction where Vt is being calculated as opposed to using Vt to pull out mass. I would like to suggest that the authors should consider the Heymsfield and Westbrook (2010) relationship. In that publication, they showed that a small modification to the Re-X relationship led to much better Vt calculations. The adjustment was a factor of the square root of the area ratio. It should be simple to determine the area ratio for the particles in the dataset and test this. Would the use of this modification improve the results (tighten the scatter)?

*Response:*
*Thank you for suggesting the modified Re-X relationship by Heymsfield and Westbrook 2010 (H&W2010). It is indeed easy to determine the area ratio in our dataset, and it is already included as a parameter. We have therefore added the modified Best number X\* as suggested by H&W2010 to our*

*methodology and recalculated mass for all particles. This results effectively in scaling up mass by a factor of $1/Ar^{0.5}$. This correction factor ranges between 1.0 and 3.8 (median 1.5). It is largest for the particles with the lowest area ratios Ar. The results improve in terms of increased correlation (higher R2) and slightly smaller uncertainties in the fit parameters aD, bD, aA, bA, am, and bm. The slopes bD are higher for most shape groups due to the general size dependence of Ar resulting in an increasing correction factor with increasing size. The changes are most noticeable in shape groups (1)-(3), which have the lowest area ratios. Both values for bD and R2 increase considerably, for example for shape group (3) from bD= 0.81 ± 0.33 (R2=0.51) to bD= 1.13 ± 0.28 (R2=0.73).*

**Changes to the manuscript:**
When revising the manuscript, the following changes will be done:
**Method section:** introduce/explain the modified X approach by H&W2010.
**Tables:** update values.
**Figures:** update.
**Discussion of $R^2$s and slopes:** adjust.

**2) The dataset also presents an interesting opportunity for manual comparisons, which would also assist in the selection of the traditional Re-X relationship or the Heymsfield and Westbrook modification. The dataset is said to contain 317 needles and 103 "thick columns". I expect that for at least a portion of these, it should be possible to geometrically estimate the mass. These particles will have the lowest area ratio values and thus would be impacted the most by the Heymsfield and Westbrook modification. It would be a great addition to the manuscript to include a small closure study between Vt, mass, and area with the Re-X relationship, either the traditional relationship or the modified relationship.**

*Response:*
*Thank you for suggestion, it is an interesting case study indeed. Our shape group (1) Needles contains many bundles of needles and only few pristine needles yet. Shape group (3) Thick columns, on the other hand, contains many simple columns. With these we have tested the closure you are suggesting. We have determined manually the width and the length of 75 columns out of the 103 particles in shape group (3). Most columns fall horizontally so that width and length can be easily determined from the top-view images. We estimate that the length may be underestimated on the order of up 15% due to deviation from alignment of the column axis in the image plane. On the other hand, the geometrically determined mass, $m_{geom}$, may be overestimated for part of the columns that show signs of cavities or hollowing of faces.*

*From width and length we have determined a characteristic length L\* as defined by Jayaweera 1971, JAS (and suggested by referee 1, David Mitchell.*

*Now, Re can be determined from measured fall speed and L\*. The Best number can be determined from measured cross-sectional area A and the characteristic length L\*. Thus, X-Re can be plotted and compared to the X-Re relationship (Eq. 5). The following figure shows the results:*

[Figure]

*Points are shown for both X-Re and modified X\*-Re where $X^* = X \cdot Ar^{0.5}$ and X calculated from Eq 3. As can be seen, the modified X\*-Re agrees better with the X-Re relationship (Eq 5). The figure also shows X-Re where Dmax was used instead of the characteristic length L\*. It shows that L\* is indeed more suitable to determine Re and X as the points determined using Dmax are further away from the X-Re relationship (Eq 5). As we now have three different sets of parameters used in Eq5, all three have been used to plot the X-Re relationship. They differ by at most about a factor 1.5 at Re = 30 and by less for lower Re. Thus, the same conclusions can be drawn regardless of the chosen parameters. We can also examine mass vs size. The following figure compares mass as determined in our manuscript and geometrically calculated mass versus Dmax or characteristic length L\*:*

[Figure]

*Circles and triangles show mass of the 75 columns from shape group (3) against Dmax. The lowest mass is that determined as described in Sect 3.1 and Sect. 3.2 in the manuscript. The mass using modified X\* is larger by the factor 1/Ar$^{0.5}$ (as explained in our response to your general comment 1) but has apparently a similar spread. Mass determined using characteristic length L\* instead of Dmax in Eq.s 2 and 6 is yet larger and it is somewhat better correlated (R$^2$=0.57 vs 0.44 an 0.21 for mass from modified X\* and mass as in the manuscript, respectively). The mass geometrically calculated from width and length is again larger. It is also better correlated to a power law (R$^2$=0.69) and has a steeper slope (bD=1.52, as reported in the legend). It is also most similar to the relationship by M96 for hexagonal columns (also shown in the figure). If mass cannot be determined geometrically, then either mass using L\* or modified X\* is a better estimate of mass than mass using the unmodified X as described so far in our manuscript.*

*Mass vs characteristic length is also shown in the figure. Both mass determined by the method of our manuscript, but using characteristic length L\* instead of Dmax in Eq.s 2 and 6, and mass calculated geometrically have slopes of the fitted power laws of close to 3. Mass using L\* is lower but comes within a factor of 1.5 to the geometrically calculated mass.*

**Changes to the manuscript:**
We are considering adding these results and the discussion to the Appendix.

**3) The authors go to great efforts to present results by different shape categories. As they point out though, remote sensing cannot identify the microphysical characteristics of particles, so shouldn't we be emphasizing overall characteristics of the particle populations? Rather than emphasizing the differences in characteristics between different particle categories, I'd suggest showing more "average values" and characterize the uncertainty that could exist. I seriously doubt that weather modeling will get to the point where we can predict 15 different particle categories, so understanding the intimate details of each isn't going to be too critical too soon.**

*Response:*
*We understand your concerns considering that models unlikely will incorporate 15 different shape categories any time soon. Sorting our data into 15 shape groups is an attempt to use the shape information that we have to describe how particle properties depend on shape. While we do not think that these 15 groups are ultimately best suited to do that, it has been a starting point for our studies Vázquez-Martín et al. 2020 Appl. Sci. and Vázquez-Martín et al. 2021 ACP. Analysing differences between different shapes will help to characterize ranges and uncertainties related to greatly varying snow particle properties. Thus, it may also be useful in case models want to include shape dependencies starting with only two or three shape categories.*

■ **Other comments:**

**1) The "Dataset" section could have an additional paragraph with a brief description / summarization of how the dataset was created. While the reader can obviously go to the two Vázquez-Martín et al publications for more information, a one paragraph summary could potentially save the reader some time.**

*Response:*
*Thank you for pointing this out. We will add your suggestions to the "Dataset" section.*
**Changes to the manuscript:**
- Lines 59-60, between "The dataset consists of 2461 … and other hydrometeors" and "The data have been collected using D-ICI …" ADDED a new sentence:

"The dataset consists of 2461 … and other hydrometeors. Same dataset have been used in Vázquez-Martín et al. (2020b). The data have been collected using D-ICI …"
- Lines 61-62, between sentences "… the winters of 2014/2015 to 2018/2019" and "Snow particles are imaged simultaneously from …" ADDED a new sentence:

"… the winters of 2014/2015 to 2018/2019. The images are determined when the snow particles fall into the inlet and consequently fall down the sampling tube and traverse the optical cell. In the centre of the optical cell is the sensing volume. If particles are falling through the sensing volume they are detected by the detecting optics (detailed description in Kuhn and Vázquez-Martín (2020). Upon detection, the particles are optically  imaged simultaneously from two different viewing directions."

**2) Fitting the dataset: I applaud the appropriate use of data. Reasonable binning and using the median values is something that eludes many. Good job.**

*Response:*
*Thank you for your positive feedback.*

**3) In general, it might be nice to have a table with the symbols / variables used. You use a lot of sub scripts and super scripts and tildes etc. While it is easy to follow what each means after reading it a few times, it took me a few times reading the equations to completely follow the variables.**

*Response:*
*Thank you for your suggestion, we are considering to add a list of symbols/variables and acronyms used as a new appendix.*
**Changes to the manuscript:**
Appendix C: List of Symbols, Variables, and Acronyms.

**4) Table 1: Some of your shape categories had as few as 37 particles. When you separated into size bins, did you have a minimum number of particles per bin? (Did you do six bins of 6-7 particles or 10 bins of 3-4 particles?) You might consider noting the temperature range that particles in each type were observed if that were possible, and, if possible, the number of days when observations of each particle type were observed. 5 winters of data may not seem biased, but if all 37 graupel particles were observed on one day, then there could be a bias.**

*Response:*
*As we have explained in Vázquez-Martín et al. (2020b ACPD, now 2021 ACP), the data are first binned into 10 particle size or cross-sectional area bins before fitting to Eqs. (11) and (12), respectively, where the binning uses flexible bin widths so that each bin contains as close to the same number of particles as possible. Therefore, the bin widths are different for each shape group and thereby avoid the problem of individual bins having a disproportional effect on the fit. The number of bins (10) is a compromise to a small enough number of bins to contain enough particles per bin and a large enough number of bins to allow for a good fit to the measurements. So, as you have pointed out, for a shape group with only few particles, such as (12) graupel with 37, only few particles are included in each bin, such as 3-4 for graupel. We found about 40 particles to be a limit where the binning can still be used. However, at such low particle numbers, the binning method is not better than fitting to individual data.*
*In most shape groups, particles come from a number of different days and winters. In Graupel for example, particles had been collected on 12 days in 3 winters. Therefore, the risk for potential biases due to sampling under very few special conditions is reduced.*
*We could not find clear relations between our measured ground temperature and shape groups. Temperature ranges were in general broad. While this at first may seem unexpected, it is understandable considering that the temperature on the ground does not have a unique relation to the conditions at formation and thus is not directly related to snow habit. We decide therefore to not report these ground temperature ranges since they alone did not convey any useful information.*

**Changes to the manuscript:**
**Add to the end of Sect. 3.2, Line 125:**
Vázquez-Martín et al. (2021 ACP) found that about 40 particles in a shape group (currently the lowest number in our dataset is 37) is the limit where binning can still be used. The advantages of binning, however, become prominent only at larger numbers of particles.

**5) Figure 1: The "fit to all data" appears to have a tighter uncertainty range. Also, the size range doesn't go as far as one might expect. Again, statistically, how far off would a user be if they used an average value and the particles were mostly one of the categories?**

*Response:*
*The uncertainty range of the fitted power law, i.e. the 68% confidence region of the fit, is indeed more narrow than for most of the shape groups. The "fit to all data" is the fitted power law to all 2461 particles. This large number is likely the main reason for the relatively narrow confidence region. Some shape groups, however, do have a narrower range, such as for example for m-Dmax of groups (8) Branches and (9) side planes, which have the highest number of particles among the shape groups. Thus, "all data", being a mix of different shapes, has a wider confidence region as these shapes with "good statistics". One should also look at the prediction band corresponding to the fits without using data binning, i.e. the range given by the expected 16th and 84th percentiles of mass (where the majority of new measurements would be expected). To avoid confusion with too many lines, we have decided to not show these regions in Fig. 1. In Fig 1a, this prediction band for "all data" would span over about a factor of 4 in mass.*
*The length of any of the fit lines, including the line for all data, is defined by the percentiles 16th and 84th of all values of Dmax (m vs Dmax), of A (m vs A), and of m (v vs m). Thus, the length indicates the Dmax, A, or m regions where most data are found. Had we shown the percentiles 0 and 100 then the regions for all data would encompass all regions of all shape groups.*

**6) Figure 2: I'd suggest removing this figure as it doesn't seem important.**

*Response*
*This figure is not essential. The related discussion can be shortened without referring to it while still remaining clear. We will remove the figure and make the necessary modifications to the text.*
**Changes to the manuscript:**
**Remove Fig. 2**
**Lines 165-169:**
"Figure 2 compares the coefficients RD2 and RA2 of all the shape groups, and for most shape groups, the two are similar. Only the four groups (1), (2), (3), and (10), mentioned above with lower correlation in one of the relationships, have a distinct difference between RD2 and RA2 . These are clearly above and below the line representing RA2 = RD2 in Fig. 2. Of these, the three shape groups (1)–(3) above the line have a better correlation for m vs A than for m vs Dmax, which is consistent…"
CHANGE TO:
"In most shape groups, the coefficients RD2 and RA2 are similar. Only the four groups (1), (2), (3), and (10), mentioned above with lower correlation in one of the relationships, have a distinct difference

between RD2 and RA2 . Of these, the three shape groups (1)–(3) have a better correlation for m vs A than for m vs Dmax, which is consistent…"

**Line 309:**

"…similar to the corresponding values of RA2 (see Fig. 2)."

CHANGE TO:

"…similar to the corresponding values of RA2."

**7) Figure 3: It would be interesting to look at the ratio of the powers in the mass to D relationship versus the Area to D relationship. Mitchell 1996 has many comparisons and Schmitt and Heymsfield 2010 show that this ratio would be ~1.3 for fractal particles and of course, it would be 1.5 for spheres.**

*Response*
*Thank you for this suggestion. Looking at the ratios of the powers in the m-Dmax relationship versus the A-Dmax relationship reveals some problematic shape groups where the power exponent in the m-Dmax relationship, bD, is somewhat smaller than the power exponent in A-Dmax, b. It was not surprising to find groups (1)-(3) among the offending shape groups. For example, in group (3) bD=1.1 and b=1.2. The only other two groups with this problem are (9) Side planes and (10) Spatial plates. Group (10) has few particles and was noticed before for bad correlations between fall speed and any of Dmax or A. For group (9) bD=1.4 and b=1.8. This problem likely indicates for which shape groups Dmax is not suitable as size parameter to calculate Re. Then, for these shapes or shape groups one should use a suitably defined characteristic length in Eqs. 2 and 6. As mentioned earlier, currently we cannot easily determine characteristic length, as we have done for some regular columns, for all particles. Thus, at the moment the modified X\* approach according to H&W2010 remains the best for our study, it lessens the problem considerably for groups (1)-(3).*

**Changes to the manuscript:**

**Lines 206-207:**

"Instead, the X–Re relationship given by Eq. 5 not describing well this shape group may be responsible again.

CHANGE TO:

"<new paragraph>

Shape group (9) is also noticed when comparing bd, the exponent of the m-Dmax relationship, to b, the exponent of A-Dmax. Intuitively, bD should be larger than as also confirmed by literature, such as by M96. For some shape groups, however, b is larger than bD. Not surprisingly, groups (1)-(3) that were noticed earlier by having the lowest bD values are among these groups, as well as groups (9) and (10). The latter was noticed in Vazquez-Martin (2021 ACP) with very poor correlations between Dmax or A and fall speed. This problem likely indicates that for these shape groups Dmax is not suitable as size parameter to calculate Re. While suitable substitutes exist for regular shapes, such as the characteristic length suggested by Jayaweera (1971, JAS), for an arbitrary shape our current image analysis methods cannot determine a similar quantity. Thus, the modified X\* approach according to H&W2010 remains the best alternative for our study, it lessens the problem considerably for groups (1)-(3).

**Summary and conclusions:**

Add another bullet point about this issue of Dmax not being well suited to calculate Re for some shapes.

---

## Author Response (AR1)

David Mitchell – Referee #1

**Received and published: 23 April 2021**

**Dear David Mitchell,**

We thank you for your constructive feedback. We sincerely appreciate that you took the time to read and evaluate our work. Please see below our response to your comments.

We are reporting our response (in blue italics) directly following each point that you have raised. Then, we are suggesting changes to the manuscript (still in blue).

**■ General comments:**

This paper is the first to demonstrate the calculation of ice particle mass from measurements of ice particle maximum dimension, projected area and fall velocity, and in doing so, it represents a test of hydrodynamic flow theory. However, as argued below, the method appears successful for graupel and quasi-spherical ice particles, but less successful for planar ice crystals (e.g., stellars or dendrites) and definitely not successful for needles and columnar ice crystals. Similar findings were reported by Heymsfield and Westbrook (2010, JAS; henceforth HW2010), where the fall speed treatment of Mitchell (1996, JAS; henceforth M96) worked well for ice crystals having aspect ratios closer to unity (e.g., graupel, short columns, thick plates) but not well for stellars and needles having more extreme aspect ratios. Therefore, two new approaches are offered for modifying the methodology described in this study.

One approach is to define the Reynolds number Re in terms of a characteristic length L\*, rather than maximum dimension Dmax, where  $\text{Re}_{L^*} = V L^*/v$ , where V = terminal fall speed and v = kinematic viscosity of air =  $\eta/\rho_a$  where  $\rho a =$  air density and  $\eta =$  dynamic viscosity of air. This L\* was found to describe the vapor mass and heat transfer from a ventilated ice crystal well and hence captures the flow effect on ice crystal growth. L\* is defined as

**$L^{*} = A/P(1)$**

where A = total surface area of an ice particle and P = ice particle perimeter projected to the flow. See Pruppacher and Klett (1997), *Microphysics of Clouds and Precipitation, Kluwer Academic Publishers*, p. 552, for more details.

Fortunately, Jayaweera (1971, JAS) has formulas describing L\* for planar and columnar ice crystals. For planar ice crystals,  $L^* = (d/2) (1 + 2e)$  and

(2)  $\operatorname{Re}_{L^*} = 0.5(1 + 2e) \operatorname{Re}_d$

where subscript d on Reynolds number Re indicates that Re is evaluated using the diameter d of a circle having the same area as the basal face of the ice crystal. For hexagonal plates, d = 0.910 D, where D = maximum dimension of the basal face. Moreover, e refers to the ratio of minor to major axis. Reasonable estimates for e can be obtained from Dmax and Auer and Veal (1970, JAS). For columnar ice crystals, L\* = ( $\pi/4$ )d [1 + (1/(1+e))] and

(3)  $\operatorname{Re}_{L^*} = (\pi/4) [1 + (1/(1+e))] \operatorname{Re}_d$ .

Notice here that  $Re_{L^*}$  depends on the "diameter" or thickness of a column and not its maximum dimension Dmax (as was used in this ACPD paper). This indicates V is most related to d. By substituting ReDmax used in this ACPD paper with  $Re_{L^*}$  in Eqn. 5 to calculate Best number "X", and substituting Dmax with L\* in Eqn. 6 of this paper (while using this new calculation for X), the m-Dmax power laws found here for columnar and planar ice crystals may be improved, having greater consistency with the body of theoretical and empirical knowledge (discussed below under Major Comments).

The second approach is to calculate the "modified Best number" X\* from Eqn. 5 of this ACPD paper (as described in HW2010) by redefining the constants used to calculate X\*, where C0 = 0.35 and  $\delta0 = 8.0$ . However, in this case Re = ReDmax (as originally used in this paper) since this maintains consistency with HW2010. Then mass m is calculated by inverting the HW2010 definition of X\*:

 $m = \pi \eta^2 X^* Ar^{1/2} / (8 g \rho_a) (4)$

where m = ice particle mass, g = gravity constant and Ar = area of ice crystal normal to the flow divided by area of circle having same maximum dimension (referred to as the area ratio).

However, if the "thick columns" shape category in this study corresponds to short, thick columns, the M96 fall speed scheme should work fine for this shape category, and this second approach may not address the problem for this shape.

It is possible that neither of these alternative approaches will render improved results, but it seems worth a try. If there is no improvement, the limitations described below will need to be mentioned in the paper.

**Response:**

Thank you very much for suggesting these two approaches to modify and improve our methodology. The first approach seems to have much potential for better describing falling snow particles. Unfortunately, it is not easy to determine and add a characteristic length to all particles. This may be attempted after developing more automatic methods to retrieve particle properties from the image data. For a subset of shape group (3) we have performed a manual analysis to determine widths and lengths of hexagonal columns. From these dimensions, the characteristic length as defined by the study Jayaweera (1971, JAS) that you mentioned can be calculated and then used to determine Re and m. The results of this case study will be shown and discussed in our responses to RC2 general comment 1, as referee 2 suggested such a case study to look at the X-Re relationship since m can be estimated directly from these dimensions without using X-Re.

For our current ACPD study, the second approach that you have suggested, i.e. using a modified Best number X\* as proposed by Heymsfield and Westbrook 2010 (H&W2010), can be easily added to our methodology as the needed area ratio is already part of our dataset. We are therefore implementing this, for more details please see our response to general comment from RC2.

**Major comments:**

1) Lines 149-150: Please indicate which relationships in Tables 1 and 2 are based on the approach described in Sect. 3.3 vs. the approach given in Sect. 3.2. Since this approach involves empirical relationships already having considerable uncertainty, m(D), m(A) and v(m) from this approach may have greater uncertainty than the previous approach (Sect. 3.2) due to the propagation of uncertainties in the empirical expressions used here. This knowledge may be helpful for interpreting the results in Tables 1 and 2.

**Response:**

The mass of each particle in our dataset is calculated based on measured maximum dimension, crosssectional area, and fall speed as described in Sect. 3.1. All the relationships that we report in Table 1 are based on the fitting method described in Sect. 3.2. Table 2 contains both our results and previously reported results. Our results are the same as in Table 1 and are repeated for convenience. The previous results in Table 2 are shown as reported in literature, i.e., we did not use any of the analytical relationships described in Sect. 3.3 to convert any of those previous relationships. The derived relationships are reported in Sect. 3.3 for convenience in the case that a suitable dataset is not available, but existing relationships are. With our dataset we have the choice to use either method. By using the method outlined in Sects 3.1 and 3.2 we avoid issues with error propagation. We have clarified that in the manuscript.

**Changes to the manuscript:**

Line 149-150: "Note, that both methods for deriving the relationships given by Eq. 7–Eq. 9, described in Sect. 3.1 and in this section, are equivalent if they are based on the same dataset."

CHANGE TO: "Note, that both methods for deriving the relationships given by Eq. 7–Eq. 9, that is either the method described in Sect. 3.1 with fitting detailed in Sect. 3.2 or the alternative derivation from existing relationships described in this section, are equivalent if they are based on the same dataset."

Add at end of Sect. 3.3: "Thus, in this study, we have chosen to fit our data directly to Eq. 7–Eq. 9 (Sect. 3.2). This allows using environmental conditions individually for each particle and avoids the need to consider error propagation when deriving new relationships from existing ones."

Line 244: "Relationships from this work are further referred to as [VM]."

CHANGE TO: "Relationships from this work are further referred to as [VM] and are taken from Table 1. They have been determined as described in Sect. 3.2."

**Caption Table 2:** "The relationships found in this work are also shown as [VM]." CHANGE TO:

"The relationships in this work have been found by fitting our dataset to Eq. 7–Eq. 9 as described in Sect. 3.2 and reported in Table 1. Those selected for comparison are also shown here as [VM]."

2) Lines 185-188 on stellar ice crystals: This same argument is expressed more quantitatively in Mitchell et al. (1990, Sect. 4a), where hexagonal crystal volume V is approximated by using a circular basal face so that aspect ratio k = c/a = c/r, where r = radius of this circle having the same area as the basal face. Moreover, c and a are the semi-axes corresponding to the prism and basal faces of an ice crystal. Thus,  $V = \pi r^2 k r = \pi k r^3$  for this approximation, and for constant density and constant k, the m-D relationship has a power of 3.

However, constant k is an invalid assumption for a stellar ice crystal that only forms between ~-14 and -16 °C. As described in Chen and Lamb (1994, JAS), k is rarely constant (definitely not at these temperatures), and depends on the ratio of condensation coefficients for the basal and prism crystal faces. This is referred to as the inherent growth ratio (IGR), and IGR is related to m-D power laws in Sect. 7 of their paper. I strongly recommend that the authors read this paper and then revise this commentary accordingly. This information is also in the cloud physics text book by Lamb and Verlinde, Physics and Chemistry of Clouds (2011, Cambridge Univ. Press). In addition, Jerry Harrington's group at Penn. State Univ. has greatly extended this work through several publications (e.g., Harrington et al., 2013, JAS, "A method for adaptive habit prediction in bulk microphysical models. Part I: Theoretical development").

Physical intuition also informs us that slope bD is too high since bD is a measure of the increase in mass with respect to size. A large (i.e., steep) slope indicates a relatively large mass increase per unit size increment, but this is not true for stellar or dendrite ice crystals since their ice density decreases with increasing size.

**Response:**

The argument presented in Mitchel et al. 1990 Sect 4a applies to plates or columns with constant aspect ratio. While, presented in this way, it is very clear, we chose to express it in a more general way since we were referring to shape groups 6 stellar and 11 spatial stellar. For stellar particles one would, as you are pointing out, not expect a high exponent (steep relationship) as we are observing for shape groups 6 and 11. Instead you would expect a decreasing density, or area ratio, with increasing size. However, in Vazquez Martin et al. 2020 (Figure 5 Bottom) we showed that, unexpectedly, the area ratio in this group is almost constant. This may be caused by a particular mix of shapes in shape group (6). Besides pristine stellar shapes, the group contains other shapes, such as for example rimed stellar and split stellar crystals. These two shapes (rimed stellar and split stellar crystals) account for 15 out of 43 particles in shape group (6) and excluding them results in a reduced exponent (less steep slope). These numbers, however, also highlight that the shape group suffers from bad statistics due to a low number of particles. Thus, it will be interesting to revisit this issue later, when more pristine stellar particles will have been added to our dataset. We will highlight these issues more clearly in the manuscript.

**Changes to the manuscript: Line 187-188:**

"However, a slope similar to spherical particles indicates that in these groups the morphology remains similar independent of size, i.e. during growth the ice particles grow equally in all three dimensions." CHANGE TO: "However, a slope similar to spherical particles may indicate that in these groups the morphology remains similar independent of size, i.e. ice particles scale equally in all three dimensions. An example for this would be hexagonal plates or columns that all have the same aspect ratio. For pristine stellar particles one may not expect such a steep slope similar to spherical particles, but rather a decreasing area ratio with increasing size. Shape group (6), however, contains other shapes besides pristine stellar particles, such as rimed stellar and split stellar crystals. A particular mix of shapes may cause an apparently steep slope. Indeed, the area ratio in this shape group is approximately constant (Vazquez Martin et al., 2020). Our dataset does not contain a sufficient number of stellar particles yet to analyse this further, by for example regrouping particle shapes. Additionally, the low number of particles in this group also results in a relatively high uncertainty ( $b_{d} = 2.60 \pm 0.69$ )."

3) Lines 190-196: These shapes are all columnar, which may be a clue to the problem here. Reynolds Number Re is expressed by (2) in terms of  $D_{max}$ , but due to hydrodynamical considerations, some argue that Dmax should be replaced by a "characteristic length" L\* defined in Pruppacher and Klett (1997) as: L\* = A/P, where A = total ice particle surface area and P = particle perimeter normal to the flow. For needles, changes in A and P will be roughly proportional, with L\* changing much less than  $D_{max}$ .

As shown by Jayaweera (1971, JAS), L\* is strongly related to the column radius (or basal face semi-axis) and weakly related to Dmax, indicating the formulation of Eqns. (2) and (5) in terms of Dmax are flawed based on ReL\*. This is indeed the case for hexagonal columns.

**For our response, see Response to your major comment 4)**

4) Lines 196-199: Note that  $b_D < 1.0$  also for thick columns (C1e; group 3), with  $b_D = 0.81$ . Compare this with Tables 1 & 3 in Mitchell et al. (1990, JAM)), where C1e  $b_D = 6$  is found to be consistent with other studies based on dimensional-density relationships. Moreover,  $b_D = 2.6$  is very consistent with the theoretical prediction of Chen and Lamb for C1e  $b_D$  (see their Fig. 12). This is strong evidence that the C1e  $b_D$  in this current study suffers from some limitation. Please expose this issue for the readers.

**Response (to major comments 3) and 4) together):**

In Vazquez Martin 2021 (ACP) we showed that fall speed is very poorly correlated to Dmax for shape groups (1) Needles, (2) Crossed Needles, and (3) Thick columns. This supports the point that you are highlighting, i.e. that Dmax is not suitable to describe relationships. As we have described there, and as you are pointing out, a characteristic length that is similar to the width of columnar particles would be more suitable as it should be used to determine Reynolds number. Since we are using Reynolds number (together with the X-Re relationship), this has consequences for the resulting mass. For these three shape groups, related to columnar particles. Using Dmax will result in unreliable and/or incorrect mass. As you have suggested, replacing Dmax with the width (diameter) and using Re based

on that, should result in a better relationship between size (now width) and derived mass. Indeed, for a subset of 75 particles in shape group (3), for which column width can be easily defined,  $b_D$  is 2.4. This seems consistent with  $b_D = 2.6$  by Mitchell et al. 1990 (Tab. 1), who, however, have used column length (if we interpreted L in that paper correctly) rather than width.

We will include these arguments to improve the discussion and conclusions.

**Changes to the manuscript:**

**Line 192-199:**

"We have seen in Vázquez-Martín et al. (2020b) that an increase in Dmax (needle length) is directly proportional to A, indicating that the diameter of these needle-shaped particles (needle width) remains similar, whereas Dmax, and consequently A is growing. Thus, these shapes are clear examples of a size-dependent morphology, i.e. as size increases, not all three dimensions grow at the same rate. In this case, since Dmax is approximately proportional to A, one would expect both values of ~bD and  $bA \simeq 1$ , which most of them are for these three shape groups. Only bD for shape groups (1) and (2) are smaller than 1, indicating a decreasing width as the particle length increases. This seems inconsistent, which might be due to the X-Re relationship given by Eq. 5 not being accurate for these shapes. However, this may also be related to the very low correlation in these two cases." CHANGE TO:

"We have seen in Vázquez-Martín et al. (2020b) that an increase in Dmax (needle length) is directly proportional to A, indicating that the diameter of these needle-shaped particles (needle width) remains similar, when Dmax and consequently also A are growing. Thus, Dmax is approximately proportional to A, and one would expect both values of bD and bA to be close to 1, which most of them are for these three shape groups. Vázquez-Martín et al. (2020b), observing the very poor correlation between Dmax and measured fall speed, argued that Dmax is not suitable to determine the Reynolds number. Therefore, a more suitable characteristic length should be used, rather than Dmax, to determine Reynolds number and derive mass from it. Otherwise, the derived mass, and consequently bD, are likely not useful. Jayaweera (1971, JAS) suggested a characteristic length for hexagonal crystals for which the dimensions of the basal facet and the aspect ratio are known. Unfortunately, this information is not readily available for all particles in our dataset (or is not defined in case of more complex particles). Therefore, determined mass and relationships based on it should not be used."

**Conclusions:**

Add a disclaimer about groups (1)-(3) to first bullet point or add a new second bullet point about that: "For the three shape groups related to columnar or elongated shapes, i.e. shape groups (1)–(3), width rather than length or Dmax is more closely related to a suitable characteristic length to determine Re. Consequently, mass and relationships with it are not reliable. For these shape groups, bD is close to or smaller than 1. In addition, bD is larger than b, contrarily as expected, and the ratio of exponents bD to bA is too low for these groups. For most other shape groups it is similar to b, as theoretically expected. Shape groups (9) and (10) (the latter with low number of particles and low correlations in relationships) show similar limitations when comparing with b. Therefore, as long as a more suitable size parameter is not available in our dataset for these shapes, mass derived from Re for these shape groups should only be used with great caution."

5) Section 4.4.1 on plates: Chen and Lamb (1992) provide theoretical limits for columnar and planar ice crystals regarding bD, where bD for hexagonal plates lies between 2.0 and 3.0. In this current study for plates,  $b_{\rm p} = 1.72$ , indicating its value should be treated with caution; please make readers aware of this. Since side planes grow diffusionally through a different mechanism

**(Furakawa, 1982, J. Meteor. Soc. Japan), it is not clear whether these limits apply to side planes. Please mention this.**

**Response:**

The shape group (5) Plates contains planar and plate-like shapes such as simple hexagonal plates but also stellar plates, rimed plates, split plates, and double plates. This mixture, and in particular the inclusion of shapes other than pristine plates, is likely responsible for bD in this study being below 2, which is the theoretical limit for planar particles approximated by spheroids (Chen and Lamb, 1994). While we only have a handful simple hexagonal plates, rimed, skeletal, and double plates are represented with 39 or more particles, and for these shapes bD is 1.9 for skeletal plates and 2.1 for the other two shapes. We will discuss this better in Sect 4.4.1.

**Changes to the manuscript:**

**Add to the end of Sect 4.4.1:**

"Our relationship [VM] for shape group (5) has a lower slope bD than any of the other relationships from previous studies. Chen and Lamb (1992) approximated hexagonal plates with spheroids and found a theoretical lower limit of 2 for bD of plates, which bD = 1.76 of [VM] seems to violate. While the selected previous studies with bD values larger than 2 looked at particular shapes, [VM]'s shape group (5) represents a mixture of plate-like shapes such as rimed plates, split plates, and double plates. Two of the shapes are represented with more than 40 particles, namely rimed plates (R1c) and double plates (P1o), sufficient to determine their own relationships. Both have steeper relationships with bD of approximately 2.1. Double plates are composed of two plates with a small gap in between, so that they resemble almost thicker plates. They are most similar to the thick plates (C1h) by [H] within they their size range. Most rimed plates in our dataset are thinner plates with light to moderate riming. They are most similar to hexagonal plates by [M]."

**Fig. 4a and Tab. 2:**

We are adding two relationships with the labels 1R. and 1P. In Tab. 2 they will be inserted after 1. [VM], in Fig 4a as black dashed lines:

| Rimed plates                      | 44                | 0.370.6 mm | 1.217 μg | 0.110.6 m s -1 | 1R. m /(µg)       |
|-----------------------------------|-------------------|------------|----------|---------------------------|-------------------|
| $= 21.1 \cdot (D/mm)^{2.0}$       | 6 0.66 | [VM]       |          |                           |                   |
| Double plates                     | 55                | 0.210.9 mm | 1.758 μg | 0.110.6 m s -1 | 1P. m /( $\mu$ g) |
| $= 31.3 \cdot (D/mm)^{2.15} 0.88$ |                   | [VM]       |          |                           |                   |

**(5) Plates**

Fig 4a

6) Figure 3 caption, last sentence: The text indicates that this should be valid for all ice particles and not just spheres; please make this clear.

**Response:**

You are correct, it is valid in general regardless of shape. We will correct it in the manuscript. **Changes to the manuscript: Last sentence in caption Fig 3:** "The green solid line represents a reference for spheres, which corresponds  $\tilde{b}D/\tilde{b}A = b$ ." **CHANGE TO:** "The green solid line corresponds to the general relationship between the slopes,  $\tilde{b}D/\tilde{b}A = b$  (derived from Eq. 7, 8, and 10)."

**7) Lines 290-291: Can it be said that Re-X represents the fall speed upper limit?**

Response:

Yes, this could be said. We should then also mention that the fit line [VM] fits well our data at masses below 10 µg. The only two particles heavier than 10 µg would be overpredicted by the fit line (see Fig. A3, panel for shape group 15).

**Changes to the manuscript: Line 289-290:**

"The straight line for the shape group (15) of [VM] is at somewhat lower fall speeds below approximately 10  $\mu$ g and at higher speeds above that mass. It represents ..." CHANGE TO:

"The straight line for the shape group (15) of [VM] is at somewhat lower fall speeds below approximately 10  $\mu$ g. All data but two particles in shape group (15) have m below that mass. For those two particles heavier than 10  $\mu$ g the fit line [VM] overpredicts mass (see Fig. A3 in the Appendix). While [VM] represents ..."

**Line 291:**

"..., whereas the two curved lines of [G] and [Re–X] represent only liquid droplets." CHANGE TO:

"..., the two curved lines of [G] and [Re–X] represent only liquid droplets, and, thus, an upper limit in fall speed."

8) Summary and conclusions; 1st bullet: Will the fact that m is derived from v produce a covariance that contributes to the stronger correlation between v and m? If so, please mention this wherever it is most appropriate.

**Response:**

*Yes, thank you for pointing this out. We have already mentioned this in the last sentence in Sect. 4.1. We will repeat a similar statement in the Summary and conclusions.*

**Changes to the manuscript:**

**Line 310, add a last sentence to this bullet point:**

"The fact that m is derived from v contributes to a stronger correlation between both quantities."

9) Summary and conclusions; 2nd bullet: Will not the power-law approximations have greater uncertainty than the relationships based on Eq. 5? Please address this concern wherever it is most appropriate.

**Response:**

This point is related to your major comment 1), so please also look at our response to that comment. If the same data is used to derive first a v-D relationship and then a m-D relationship from that, or to derive a m-D relationship directly m values that have been added to the data (Sect 3.1), then the resulting m-D are identical (as long as the same X-Re relationship is used. With our dataset we have the choice to use either method. By deriving m-D directly (the method outlined in Sects 3.1 and 3.2) we avoid having to consider error propagation, we can use Eq 5 (instead of a power-law approximation of Eq. 5). In addition to changes in response to your major comment 1) we will also modify the text in this bullet point to make this clearer.

**Changes to the manuscript:**

**Lines 311-314:**

"When deriving the m vs Dmax, m vs A, and v vs m relationships analytically from A vs Dmax (see Section 3.3), the results are equivalent to fitting to measured data. The analytical relationships Eq. 13– Eq. 15 can be used if power laws are available instead of data. However, fitting to data has the advantage that Eq. 5 can be used rather than power-law approximations required for the analytical derivation of relationships (see B in Appendix)."

**CHANGE TO:**

"When deriving the m vs Dmax, m vs A, and v vs m relationships analytically from A vs Dmax, v vs Dmax, and v vs A given from a suitable dataset (see Sect. 3.3), the results are equivalent to fitting to the same dataset after adding m for individual particles derived from v (See Sect. 3.1). On the one hand, fitting m vs Dmax, m vs A, and v vs m relationships to data has the advantage that the X-Re relationship from Eq. 5 can be used rather than power-law approximations required for the analytical derivation of the same relationships (see B in Appendix). On the other hand, if a suitable dataset is not available but power-law relationships for A vs Dmax, v vs Dmax, and v vs A are, the analytically derived mass relationships Eq. 13–Eq. 15 can be used."

10) Summary and conclusions; last sentence of 3rd bullet: But we know this is not true based on Chen and Lamb (1994, JAS) and other m-D measurements for stellar crystals. Please remove this last sentence.

**Response:**

Thank you for pointing this out. As the reason for these high slopes of groups (6) and (11) is uncertain (see also our response to your major comment 2), we will remove that statement.

**Changes to the manuscript:**

**Lines 315-319:**

"Their values are highest for the shape groups (6) Stellar, (11) Spatial stellar, (12) Graupel, and (15) Spherical, close to the values for spheres, i.e. bD = 3 and bA = 3/2. While this is as expected for shape groups (12) and (15), for groups (6) and (11) it indicates that the morphology in these shape groups remains similar independent of size, i.e. during growth the ice particles grow equally in all three dimensions."

**CHANGE TO:**

"Their values are highest for the shape groups (6) Stellar, (11) Spatial stellar, (12) Graupel, and (15) Spherical. For groups (12) and (15) they are close to the values expected for spheres, i.e. bD = 3 and bA = 3/2."

**11) Line 328-330: I don't see this relationship plotted (spherical ice having a density of 0.12 g cm-3). If it is not shown, then please add in parentheses, "not shown".**

**Response:**

*Thank* you, we agree. Accordingly, we will modify this conclusion as well as the corresponding sentence in the discussion (Sect. 4.4.3). **Changes to the manuscript:**

**Changes to the manuscri**

"..., and it is well approximated, by the mass of spherical particles with a density of 0.12 g cm–3. CHANGE TO:

"..., and it is well approximated by the mass of spherical particles with a density of 0.12 g cm-3 (not shown in Figure 4-d)."

**Line 275:**

"It is well approximated, by the mass of spherical particles with a density of 0.12 g cm-3, which ..." CHANGE TO:

"It is well approximated by the mass of spherical particles with a density of 0.12 g cm-3 (not shown in Figure 4-d), which ..."

**Minor comments:**

**1) Line 150: Apparent typo; Sect. 3.1 => 3.2?**

**Response:**

Thank you for pointing this out. The method is actually described in Sections 3.1 and 3.2. This line will be changed in response to your major comment 1).

Changes to the manuscript: Line 150:

CHANGE TO "...described in Sect. 3.1 with fitting detailed in Sect. 3.2..."

**2) Lines 248-9: It appears that Ma has not been defined.**

**Response:**

Thank you for finding this oversight on our side. [Ma] corresponded to a study that we are not displaying, it will be removed.

**Changes to the manuscript:**

Lines 249-249:

"For [VM], as well as for [H], [E], [K], [M], and [Ma], D corresponds to Dmax." CHANGE TO: "For [VM], as well as for [H], [E], [K], and [M], D corresponds to Dmax."

**3) Line 329: The second comma is not needed.**

**Response:**

Thank you. The comma will be removed in response to your major comment 11).

**Other changes:**

**1) We will change the numbers of equations (10)-(12) as follows:**

$$A(D_{max}) = a \cdot \left(\frac{D_{max}}{1mm}\right)^b (10\text{-a}) , D_{max}(A) = a' \cdot \left(\frac{A}{1mm^2}\right)^{b'} (10\text{-b})$$

$$v(D_{max}) = a_D \cdot \left(\frac{D_{max}}{1mm}\right)^{b_D} (11\text{-a}) , D_{max}(v) = a'_D \cdot \left(\frac{v}{1ms^{-1}}\right)^{b'_D} (11\text{-b})$$

$$v(A) = a_A \cdot \left(\frac{A}{1mm^2}\right)^{b_A} (12\text{-a}) , A(v) = a'_A \cdot \left(\frac{v}{1ms^{-1}}\right)^{b'_A} (12\text{-b})$$

**2) References used in Table 2 and Figure 4** and mentioned in the text will be modified and adapted to ACP style and for consistency with "*Shape dependence of snow crystal fall speed*, S. Vázquez-Martín et al., Atmos. Chem. Phys., 21, 7545–7565, 2021":

- Locatelli and Hobbs (1974) [Lo]  $\rightarrow$  L74
- Heymsfield and Kajikawa (1987)  $[H] \rightarrow H87$
- Kajikawa (1989) [K] → K89
- Mitchell (1996)  $[M] \rightarrow M96$
- Erfani and Mitchell (2017)  $[E] \rightarrow E17$
- The relationships found in this work will be shown as VM21 instead of as [VM].

**3) The author contributions Lines 340-342 will be modified and completed as follows:**

Author contributions. TK and SVM performed the conceptualization; TK prepared the resources and the instrumentation; SVM and TK performed the experiments and data collection; SVM and TK prepared the formal analysis; SVM and TK carried out the data curation; SVM prepared the original draft; SVM, TK and SE contributed to changes and writing during review and revisions; SVM prepared the visualization that includes tables and figures; TK and SE carried out the supervision of the research project.

**4). Line 84:**

"we get the expression that does not depend on fall speed *v*:" CHANGE TO:

"to get an expression that does not depend on fall speed v:"

5) Mass range, given for example in the Abstract, was wrong.

0.1 to 230 ug SHOULD HAVE BEEN 0.2 to 310 ug.

Now, using the modified Best number approach the range is: 0.2 to 450 ug.

6) Consistently use "fit equations to data" and not "data to equations".

E.g. Line 135:

"The relationships are determined by binning the data first, before fitting to Eq. 8–Eq. 10." CHANGE TO:

"The relationships are determined by binning the data first, before fitting Eq. 8–Eq. 10 to the binned data."

**7) Exponent**

Call exponents such as b "exponent" and not "coefficient".

**8) For consistency and correctness:**

Line 31-32: "Furthermore, it can affect active and passive microwave measurements of clouds and snowfall..."

CHANGE TO

"Furthermore, it can affect retrievals from active and passive microwave measurements of clouds and snowfall..."

**9) This work => this study**

Changed "work" to "study" for better style and consistency.

We thank you for your constructive feedback and appreciate the time you spent to read and evaluate our work. Please see below our response to your comments.

We are reporting our response (in blue italics) directly following each point that you have raised. Then, we are suggesting changes to the manuscript (still in blue).

**■ General comments:**

I have read the manuscript "Mass of different snow crystal shapes derived from fall speed measurements" by Vázquez-Martín, Kuhn, and Eliasson and I find it an interesting manuscript. The authors use a dataset of measurements of 2461 ice crystals including fall speed, area, and maximum dimension. From these data they derive particle mass using standard Reynolds – Best number approaches. The dataset is then broken down into 15 different particle habit classifications and relationships between maximum dimension and the other parameters are presented and compared to previous literature.

While it is a nice study that merits being published, there are a few shortcomings that I feel should be addressed before full acceptance.

1) The project is highly dependent on the Re – X number relationship for relating mass and terminal velocity. Traditionally, this is used in the other direction where Vt is being calculated as opposed to using Vt to pull out mass. I would like to suggest that the authors should consider the Heymsfield and Westbrook (2010) relationship. In that publication, they showed that a small modification to the Re-X relationship led to much better Vt calculations. The adjustment was a factor of the square root of the area ratio. It should be simple to determine the area ratio for the particles in the dataset and test this. Would the use of this modification improve the results (tighten the scatter)?

**Response:**

Thank you for suggesting the modified Re-X relationship by Heymsfield and Westbrook 2010 (H&W2010). It is indeed easy to determine the area ratio in our dataset, and it is already included as a parameter. We have therefore added the modified Best number X\* as suggested by H&W2010 to our

methodology and recalculated mass for all particles. This results effectively in scaling up mass by a factor of  $1/Ar^{0.5}$ . This correction factor ranges between 1.0 and 3.8 (median 1.5). It is largest for the particles with the lowest area ratios Ar. The results improve in terms of increased correlation (higher R2) and slightly smaller uncertainties in the fit parameters aD, bD, aA, bA, am, and bm. The slopes bD are higher for most shape groups due to the general size dependence of Ar resulting in an increasing correction factor with increasing size. The changes are most noticeable in shape groups (1)-(3), which have the lowest area ratios. Both values for bD and R2 increase considerably, for example for shape group (3) from  $bD = 0.81 \pm 0.33$  (R2=0.51) to  $bD = 1.13 \pm 0.28$  (R2=0.73).

**Changes to the manuscript:**

When revising the manuscript, the following changes will be done:

**Introduction section:**

**Lines 48-50:**

The same Re–X relationship may also be used to determine mass if v in addition to maximum dimension and cross-sectional area is known. Szyrmer and Zawadzki (2010) have done this to determine average v vs mass relationships from measurements of snow aggregates' fall speeds." CHANGE TO:

"Heymsfield and Westbrook (2010) suggested a shape dependent modification of the Best number based on the area ratio. With this modified Best number, they showed that the error in fall speed determined from the Re–X relationship could be reduced for particles with open geometries, i.e. particles with low area ratio.

Instead of deriving fall speed from mass, the Re–X relationship may also be used to determine mass from measured fall speed. The Reynolds number can be derived from fall speed, and then mass from X together with maximum dimension and cross-sectional area. Szyrmer and Zawadzki (2010) have done this to determine average v vs mass relationships from measurements of snow aggregates' fall speeds. Lines 51-52:

"In this study, the masses of individual particles are derived from measured maximum dimensions, cross-sectional areas, and fall speeds using the dataset from our previous study ..." CHANGE TO:

"In this study, using the Re–X relationship together with the modified Best number (Heymsfield and Westbrook, 2010), the masses of individual particles are derived from measured maximum dimensions, cross-sectional areas, and fall speeds given by the dataset of our previous study..."

Method section: introduce/explain the modified X approach by H&W2010.

**Line 90:**

"where  $\delta 0 = 5.83$  and C0 = 0.6 are unit-less constants, ..."

CHANGE TO:

"where  $\delta 0$  and C0 are unit-less constants, ..."

**After line 98 add:**

"Instead of using Eq. 4 or Eq. 5 with one set of  $\delta 0$  and C0 for all particles regardless of their shape, as proposed by Böhm (1989), Heymsfield and Westbrook (2010) suggested using a modified Best number X\*, replacing X in Eq. 4 or Eq. 5, to correct for effects due to open-geometry shapes. They proposed X =X·Ar, where Ar=A/( $\pi 4$ ·Dm2ax) is the area ratio, which is close to 1 for compact shapes and the smaller the more open the geometry is. Heymsfield and Westbrook (2010) showed that using this approach they could reduce errors of determined fall speeds associated to open-geometry particles with low area ratios. Using our data for simple thick columns in shape group (3), we could confirm that their approach is better than the approach by Böhm (1989) without modifying X (see Appendix C). Therefore, here, we are using the modified Best number X\*. Consequently, Eq.6 is modified to  $m = \dots$  (7) Note, that then the Best number determined from Eq. 5 is the modified Best number X\*. In Eq. 5, we are using  $\delta 0 = 8.0$  and C0 = 0.35 from Heymsfield and Westbrook (2010)."

**Delete Lines 99-104:**

"....C0 = 0.6 and 0 = 5.83 ... for spheres: C0 = 0.292 and 0 = 9.06..."

**Appendix B:**

Change equations to account for using modified X. Consequently update also Eq.s 13-16. **Tables:** update values.

**Figures: update.**

Discussion of R2s and slopes: update values and adapt discussion accordingly, e.g.:

**Line 161:**

"...correlation  $(0.6 \le R^2 \le 1)$ "

**CHANGE TO:**

"...correlation  $(0.9 \le R^2 \le 1)$ "

**Lines 162-164:**

"For the m vs Dmax relationship, the only exceptions to high correlations are shape groups (1) Needles and (2) Crossed needles with RD2  $\approx$  0.2 as well as (3) Thick columns with RD2 = 0.51." CHANGE TO:

"For the m vs Dmax relationship, the only exceptions to high correlations are shape groups (1) Needles, (2) Crossed needles, and (3) Thick columns, as well as (6) Stellar and (10) Spatial plates, which both have low number of particles, having all RD2  $\simeq 0.7$ ."

**Line 164:**

"For the m vs A relationship, only shape group (10) Spatial plates has a lower correlation with R2A = 0.45."

CHANGE TO:

"For the m vs A relationship, only shape groups (2) and (6), with RA2  $\simeq$  0.8, and (10) with RA2  $\simeq$  0.5, have a lower correlation."

2) The dataset also presents an interesting opportunity for manual comparisons, which would also assist in the selection of the traditional Re-X relationship or the Heymsfield and Westbrook modification. The dataset is said to contain 317 needles and 103 "thick columns". I expect that for at least a portion of these, it should be possible to geometrically estimate the mass. These particles will have the lowest area ratio values and thus would be impacted the most by the Heymsfield and Westbrook modification. It would be a great addition to the manuscript to include a small closure study between Vt, mass, and area with the Re-X relationship, either the traditional relationship or the modified relationship.

**Response:**

Thank you for suggestion, it is an interesting case study indeed. Our shape group (1) Needles contains many bundles of needles and only few pristine needles yet. Shape group (3) Thick columns, on the other hand, contains many simple columns. With these we have tested the closure you are suggesting. We have determined manually the width and the length of 75 columns out of the 103 particles in shape group (3). Most columns fall horizontally so that width and length can be easily determined from the top-view images. We estimate that the length may be underestimated on the order of up 15% due to deviation from alignment of the column axis in the image plane. On the other hand, the geometrically determined mass, mgeom, may be overestimated for part of the columns that show signs of cavities or hollowing of faces.

From width and length we have determined a characteristic length L\* as defined by Jayaweera 1971, JAS (and suggested by referee 1, David Mitchell.

Now, Re can be determined from measured fall speed and  $L^*$ . The Best number can be determined from measured cross-sectional area A and the characteristic length  $L^*$ . Thus, X-Re can be plotted and compared to the X-Re relationship (Eq. 5). The following figure shows the results:

---

## Referee Report (RR1)

Referee Report for ACP
ACPD article title: Mass of different snow crystal shapes derived from fall speed measurements
Author(s): Sandra Vázquez-Martín et al.
MS No.: acp-2021-203
MS type: Research article
2nd review

**General Comments:**

The authors have done an excellent job addressing my review comments and have advanced the science of ice particle fall velocities by adding Appendix C. Most of my comments address Appendix C, followed by a few other specific comments.

Appendix C in this paper demonstrates for the first time (as per my knowledge) that characteristic length $L^*$ is superior to the ice particle maximum dimension D for estimating ice particle fall speeds (v). Since Reynold's number Re and Best number X are related through boundary layer theory, this is expected, but this has not been experimentally demonstrated until now. For this reason, I consider Appendix C as seminal work in the field of cloud physics.

As noted in Heymsfield and Westbrook (2010; henceforth HW2010), the Best number X was derived by equating the drag and gravitational forces, where $X = C_d Re^2$, with $C_d$ = drag coefficient. The dimension employed in describing this balance of forces would seem to be related to the particle's boundary layer rather than D. But the method for calculating ice fall speeds in HW2010, as well as other ice fall speed methods, employs D rather than $L^*$. This is natural since $L^* = A/P$ where A = total surface area of particle and P = particle perimeter projected to the flow, which are both difficult to measure. Perhaps what is needed in future studies is an expression relating $L^*$ to D for each shape category (assuming A and P could be measured somehow).

Appendix C is rich in scientific knowledge that might be missed as it is written. The following clarifications/questions are offered to bring out this knowledge better:
1) The closure experiment consisting of the magenta data points and curve, along with the blue data points, follow a theoretical treatment that demonstrates the superiority of $L^*$ over D.
2) The closure experiment consisting of the green data points and curve demonstrates better closure is obtained by using $X^*$ rather than X, where $X^* = X A_r^{1/2}$, where X and Re are based on $L^*$ and $A_r$ = area ratio. There is no theoretical reason (so far) for multiplying X by $A_r^{1/2}$.
3) The HW2010 scheme was derived in terms of D; not $L^*$. Moreover, the limiting value of the pressure drag coefficient, $C_0$, is 0.35, which is not supported by lab experiments.

Perhaps a more theoretical (i.e., first principle) treatment of ice fall speeds may be possible using L* (given that correcting potential overestimates in $m_{geom}$ might produce better agreement with the Re-X (Eq. 5) relationships).

4) What happens when the green data points are based on X* calculated from $m_{geom}$, A, D and $A_r$ and Re is calculated from D and v (as done in HW2010)? Is similar closure obtained? What can this tell us about the viability of the HW2010 scheme?

**Specific Comments:**

Author's response to major comments 3 & 4 (relating to lines 220-229): This may be a minor point but is something I feel the authors should be aware of. Regarding the subset of 75 particles in shape group 3 (hexagonal columns), the authors response states that when width (column diameter) is used instead of column length (i.e., $D_{max}$), the m-D power law exponent $b_D$ is 2.4. They note this appears consistent with the $b_D$ in Mitchell et al. (1990) for hex-columns, which was 2.6, but also acknowledge the latter $b_D$ depends on $D_{max}$. However, the columns in M1990 having $b_D$ = 2.6 are short, nearly isometric hex-columns, that are much different than the columns shown in Fig. C1 (i.e., the hex-column subset used in Appendix C). M1990 also features "long columns" (Figs. 1-3) that are comparable with the columns in V-M et al. Fig. C1.

Moreover, the mass-width relationship for columns obtained by the authors; $m = a\,W^{2.4}$, where W = column width, can be converted into a mass-$D_{max}$ relationship by substituting the width-length relationship for columns from Auer & Veal (1970, JAS). From Auer & Veal, $W = 11.3\,L^{0.414}$ for L > 200 um, where L = length and units are in microns. Substituting into the V-M et al. relationship gives:
$m = a\,W^{2.4} = a\,(11.3\,L^{0.414})^{2.4} = a'\,L^{0.994}$. Thus, a quasi-linear dependence is found between m & L, similar to $b_D$ = 1.1 for columns in Table 1 of V-M et al., but this time derived from column width. This also demonstrates consistency with the Auer and Veal measurements.

Line 448: Please define L* as A/P where A = total surface area and P = particle perimeter projected to the flow.

**Technical Comments:**

Line 115: Needs a "period" at the end.

---

## Author Response (AR2)

Author's response to Referee 1, David Mitchell, 2nd review from 10 Sep 2021
ACPD article title: Mass of different snow crystal shapes derived from fall speed measurements
Author(s): Sandra Vázquez-Martín et al.
MS No.: acp-2021-203
MS type: Research article
2nd review

Dear David Mitchell,

We are pleased that you liked our responses and changes. We are thankful for your previous constructive feedback and suggestions, which together with the suggestions by Referee 2 gave us the chance to improve our manuscript considerably. Once again, we appreciate your suggestions to improve clarity and the impact of our study. Please see below our response to your comments. We are repeating your comments in normal font.

*Following your comments, we are reporting* **our responses** *(in blue italics) directly following each point that you have raised.*
Then, we are suggesting **changes to the manuscript** (still in blue).

Kind regards,

The authors

**General Comments:**

The authors have done an excellent job addressing my review comments and have advanced the science of ice particle fall velocities by adding Appendix C. Most of my comments address Appendix C, followed by a few other specific comments.

Appendix C in this paper demonstrates for the first time (as per my knowledge) that characteristic length L* is superior to the ice particle maximum dimension D for estimating ice particle fall speeds (v). Since Reynold's number Re and Best number X are related through boundary layer theory, this is expected, but this has not been experimentally demonstrated until now. For this reason, I consider Appendix C as seminal work in the field of cloud physics.

**Response:**
*Thank you for underlining the importance of our closure study that we described in Appendix C. Consequently, and to follow the suggestion by the editor, we have added a new point in the Conclusions about this and mention it also in the Abstract.*
**Changes to the manuscript:**
ADDED third bullet point in **Conclusions**:
"On a selection of 75 simple columns from shape group (3), we have done a closure study (see Appendix C) to confirm the Re–X relationship, which is central in our method (see Sect. 3.1) and used by many other studies. For this, the widths and lengths of the columns have been determined

in addition to Dmax. From these geometric dimensions, the masses of the columns have been estimated directly. Then, from each column mass, the Best number X has been determined using Eq. 3. Thus, Re and X have been determined independently and consequently compared to the X–Re relationship given by Eq. 5. This closure showed the superiority of the characteristic length L∗ (Jayaweera, 1971) over Dmax, confirming that Dmax is not suitable to calculate Re and X using Eq. 2 and Eq. 3, respectively, for columns. The closure study also showed that using the modified Best number X∗ (Heymsfield and Westbrook, 2010) instead of the Best number X improved the agreement. The best closure for our subset of simple columns was achieved when using both characteristic length L∗ and modified Best number X∗ together."

**Line 361**:
ADDED reference to Sect 4.2.1 and Appendix C.
"…width … is more closely related to a suitable characteristic length to determine Re."
CHANGED TO:
"…width … is more closely related to a suitable characteristic length to determine Re (see Sect. 4.2.1 and Appendix C)."
ADDED paragraph to **Abstract** with results of closure study:
"The resulting mass–size relationships indicate that for certain shapes, in particular columns and related shapes, maximum dimension is not suitable to describe the size of snow particles when determining the Reynolds number. Consequently, mass derived from fall speed for these shapes is not reliable. A closure study done on a selection of simple columns, for which mass is determined geometrically, shows that for this shape a characteristic length, similar to the diameter of the basal facet, is superior to the maximum dimension, which is similar to the column length, as size parameter. Using a modified Best number, the Best number reduced as a function of area ratio, resulted in even better agreement in the closure study, confirming that the modified Best number approach adopted in this study represents an improvement for columns."

**Other changes to better include Appendix C as closure study:**

**Lines 227–229**:
In Sect 4.2.1, refer more prominently to the Appendix C.
**Line 224:**
ADDED clarification:
"between Dmax and measured fall speed,…"
CHANGE TO:
"between Dmax and measured fall speed for these shape groups,…"

**Lines 450–452:**
Improve discussion of size parameter in Eq 2-3:
"Note that Dmax in Eq. 3 comes from Eq. 2, i.e. it represents the size parameter best suited to calculate Re. Thus, also for calculating X one should use the characteristic length L∗ instead of Dmax."
CHANGED TO:
"Note that here, Dmax represents the same size parameter best suited to calculate Re as in Eq. 2. Thus, not only Re should be determined from L∗ (instead of Dmax), but also X."

**Title App C:**
"Reynolds and Best numbers for simple thick column"
CHANGED TO:
"Closure study---Reynolds and Best numbers for simple thick column"
**Lines 436:**

New sentences after first sentence to better introduce closure study:
"If Reynolds and Best numbers (Re and X, respectively) can be determined independently, i.e., without using any X–Re relationship, then they can be compared with the X–Re relationship by Bohm (1989) given by Eq. **5**. Thus, it represents a closure study that can confirm the X–Re relationship, which this and other studies rely on. The following explains the method and results when applied to simple columns, a small subset of our data." <new paragraph>

As noted in Heymsfield and Westbrook (2010; henceforth HW2010), the Best number X was derived by equating the drag and gravitational forces, where $X = C_d \, Re^2$, with $C_d$ = drag coefficient. The dimension employed in describing this balance of forces would seem to be related to the particle's boundary layer rather than D.  But the method for calculating ice fall speeds in HW2010, as well as other ice fall speed methods, employs D rather than $L^*$. This is natural since $L^* = A/P$ where A = total surface area of particle and P = particle perimeter projected to the flow, which are both difficult to measure.  Perhaps what is needed in future studies is an expression relating $L^*$ to D for each shape category (assuming A and P could be measured somehow).

This is a nice suggestion for future work. The total surface area A is difficult to measure as you have pointed out. However, A could be estimated for the more regular ice crystal shapes. Then, $L^\star$ to D could be determined for each shape, given sufficiently large datasets of such regular particles.

Appendix C is rich in scientific knowledge that might be missed as it is written. The following clarifications/questions are offered to bring out this knowledge better:
   1) The closure experiment consisting of the magenta data points and curve, along with the blue data points, follow a theoretical treatment that demonstrates the superiority of $L^*$ over D.

***Response:***
*You are right that we should highlight better, that this represents a closure study that demonstrates the usefulness of $L^\star$.*
**Changes to the manuscript:**
Line 454-455:
"The points related to Dmax do not match well the empirical relationship X–Re by Böhm (1989) with $\delta 0 = 5.83$ and $C0 = 0.6$. Added period at end of sentence."
CHANGED TO:
"The points related to Dmax (blue triangles) do not match well the empirical relationship X–Re (with $\delta 0 = 5.83$ and $C0 = 0.6$) by Böhm (1989) based on a theoretical treatment by (Abraham, 465 1970)."
Line 456:
"…closer to the empirical relationship."
ADDED sentence:
"…closer to the empirical relationship. Thus, this closure experiment comparing independently determined Re and X to the X–Re relationship demonstrates the superiority of characteristic length $L^\star$ over Dmax as a particle size parameter when dealing with particle mass m."

2) The closure experiment consisting of the green data points and curve demonstrates better closure is obtained by using X* rather than X, where $X^* = X A_r^{1/2}$, where X and Re are based on $L^*$ and $A_r$ = area ratio. There is no theoretical reason (so far) for multiplying X by $A_r^{1/2}$.

*Response:*
*Thank you for suggesting a clearer discussion of the improvement when using X★ instead of X.*
**Changes to the manuscript:**
Added period at end of sentence.

3) The HW2010 scheme was derived in terms of D; not $L^*$. Moreover, the limiting value of the pressure drag coefficient, $C_0$, is 0.35, which is not supported by lab experiments. Perhaps a more theoretical (i.e., first principle) treatment of ice fall speeds may be possible using $L^*$ (given that correcting potential overestimates in $m_{geom}$ might produce better agreement with the Re-X (Eq. 5) relationships).

*Response:*
*We were also wondering over which value of C0 to use. In combination with the different value in delta0, the different value of C0 produces very similar results. Thus, the changed value of C0 was perhaps not necessary?*

4) What happens when the green data points are based on X* calculated from $m_{geom}$, A, D and $A_r$ and Re is calculated from D and v (as done in HW2010)? Is similar closure obtained? What can this tell us about the viability of the HW2010 scheme?

*Response:*
*Using X★ and Dmax is better than X and Dmax.*
*Using X★ and L★ is better that X★ and Dmax.*
*From this one can probably not tell much about the HW2010. They focused on open geometries where Dmax may work better than it does for columns (or L★ would not be noticeably better than Dmax).*
*We have now shown that X★ works better than X even for columns.*
**Changes to the manuscript in response to 2)–4):**
**UPDATED** Fig. C2.
**L 457:**
"…according to X★ = X · Ar^1/2."
CHANGED TO:
"…where X★ = X · Ar^1/2 is the modified Best number suggested by Heymsfield and Westbrook (2010)."
**L 457–458:**
"The resulting points (using L∗) are also shown in Fig. C2 and are even closer to the empirical X–Re relationship."
CHANGED TO:
"The resulting points (using L∗) are also shown in Fig. C2 (green 'x') and provide an even better closure to the empirical X–Re relationship."
**EXTENDED** discussion around X★ with new sentences after L 458:
"Heymsfield and Westbrook (2010) used Dmax and not characteristic length L∗ (they focused on shapes with open geometries for which characteristic length is difficult to determine). The closure for our columns using X∗ and Dmax (cyan crosses in Fig. C2) is not as good as using X∗ and L∗,

but still better than using the unmodified Best number X and Dmax. Thus, for columns we can conclude that the modified Best number represents an improvement over the Best number. In addition, the superiority of characteristic length L∗ over Dmax for columns is given also when working with the modified Best number X∗."

**L 479:**
"Thus, the above discussion remains valid regardless of which relationship is used as comparison."
CHANGED TO:
"Thus, the above conclusions of superiority of characteristic length L★ over Dmax and improvement when using modified Best number rather than Best number remain valid regardless of which relationship is used as comparison."

**Specific Comments:**

Author's response to major comments 3 & 4 (relating to lines 220-229): This may be a minor point but is something I feel the authors should be aware of. Regarding the subset of 75 particles in shape group 3 (hexagonal columns), the authors response states that when width (column diameter) is used instead of column length (i.e., Dmax), the m-D power law exponent $b_D$ is 2.4. They note this appears consistent with the $b_D$ in Mitchell et al. (1990) for hex-columns, which was 2.6, but also acknowledge the latter $b_D$ depends on $D_{max}$. However, the columns in M1990 having $b_D$ = 2.6 are short, nearly isometric hex-columns, that are much different than the columns shown in Fig. C1 (i.e., the hex-column subset used in Appendix C). M1990 also features "long columns" (Figs. 1-3) that are comparable with the columns in V-M et al. Fig. C1.

*Response:*
*Thank you for pointing out that most (all but about the smallest ten) of our columns in the closure study are not comparable to C1e Short columns in M1990 but more to N1e Long columns.*

Moreover, the mass-width relationship for columns obtained by the authors; m = a W $^{2.4}$, where W = column width, can be converted into a mass-$D_{max}$ relationship by substituting the width-length relationship for columns from Auer & Veal (1970, JAS). From Auer & Veal, W = 11.3 L $^{0.414}$ for L > 200 um, where L = length and units are in microns. Substituting into the V-M et al. relationship gives:
m = a W $^{2.4}$ = a (11.3 L $^{0.414}$) $^{2.4}$ = a' L $^{0.994}$. Thus, a quasi-linear dependence is found between m & L, similar to $b_D$ = 1.1 for columns in Table 1 of V-M et al., but this time derived from column width. This also demonstrates consistency with the Auer and Veal measurements.

*Response:*
*Thank you for this interesting remark. By fitting only columns with L>200um we are left with 66 columns for which m = a W$^{2.2}$ => a' L$^{0.91}$. By fitting directly we get a' L$^{1.3}$, which is still close enough given the uncertainties (we get larger uncertainties when fitting W=c L$^b$).*

Line 448: Please define L* as A/P where A = total surface area and P = particle perimeter projected to the flow.

*Response:*
*As you suggest, it is worth here to give the definition rather than only referring to Pruppacher and Klett.*

**Changes to the manuscript:**
**Line 447**:
"A characteristic length L∗ (see ...)"
REPLACED with:
"A characteristic length L∗ = At/P, where At is the total surface area and P the perimeter of the particle projected to the flow (see …)"
The following two sentences are modified to improve the flow and avoid repetition.
L⋆, At, and P added to the Nomenclature.
**Line 446–447**:
In the preceding sentence, we added a reference to Sect. 4.2.1, where we discussed that Dmax is not suitable to calculate Re and mentioned characteristic length:
"… is not a suitable representative size parameter to determine Re."
CHANGE TO:
"… is not a suitable representative size parameter to determine Re,
as we have discussed in Sect. 4.2.1 and Vázquez–Martín et al. (2021)."

**Technical Comments:**

Line 115: Needs a "period" at the end.

*Response:*
*Thank you.*
**Changes to the manuscript:**
Added period at end of sentence.

---

## Author Response (AR3)

Author's response to Editor's comments to the authors from 11 Oct 2021
ACPD article title: Mass of different snow crystal shapes derived from fall speed measurements
Author(s): Sandra Vázquez-Martín et al.
MS No.: acp-2021-203
MS type: Research article

Dear Tim Garrett,

Thank you for your valuable comment. We agree that the added text in the Abstract was too laborious risking to hide its important message. You have kindly proposed a new text:

For certain crystal habits, in particular columnar shapes, the Reynolds number and fall speed are more closely related to the diameter of the basal facet than the maximum dimension. Further improvements are obtained from using a modified Best number, that is a function of an area ratio.

*We have changed the text considering your proposed new text and adding some more information trying to still stay concise. We have also included, as you suggested, a comment stating that the area ratio in this study considers a vertical viewing direction.*

**Changed text in the Abstract:**
"The resulting mass–size relationships indicate that for certain shapes, in particular columns and related shapes, maximum dimension is not suitable to describe the size of snow particles when determining the Reynolds number. Consequently, mass derived from fall speed for these shapes is not reliable. A closure study done on a selection of simple columns, for which mass is determined geometrically, shows that for this shape a characteristic length, similar to the diameter of the basal facet, is superior to the maximum dimension, which is similar to the column length, as size parameter. Using a modified Best number, the Best number reduced as a function of area ratio, resulted in even better agreement in the closure study, confirming that the modified Best number approach adopted in this study represents an improvement for columns."
REPLACED WITH:
"For certain crystal habits, in particular columnar shapes, the maximum dimension is unsuitable for determining Reynolds number. Using a selection of columns, for which the simple geometry allows the verification of an empirical Best number to Reynolds number relationship, we show that Reynolds number and fall speed are more closely related to the diameter of the basal facet than the maximum dimension. The agreement with the empirical relationship is further improved using a modified Best number, a function of an area ratio based on the falling particle seen in the vertical direction."

Kind regards,

      *The authors*